# Observation of partonic flow in proton–proton and proton–nucleus collisions

**The ALICE Collaboration** ✉*

Quantum Chromodynamics predicts a phase transition from hadronic matter to quark–gluon plasma (QGP) at high temperatures and energy densities, where quarks and gluons (partons) are no longer confined within hadrons. The QGP forms in ultrarelativistic heavy-ion collisions. Anisotropic flow coefficients, quantifying the azimuthal expansion of produced matter, probe QGP properties. Flow measurements in high-energy heavy-ion collisions show a distinctive grouping of anisotropic flow for baryons and mesons at intermediate transverse momentum – a feature associated with flow imparted at the quark level, confirming QGP existence. The observation of QGP-like features in proton–proton and proton–ion collisions has sparked debate about QGP formation in smaller systems. For the first time, we demonstrate the distinctive grouping of anisotropic flow for baryons and mesons in high-multiplicity proton–lead and proton–proton collisions at the Large Hadron Collider (LHC). These results are described by a model including hydrodynamic flow followed by hadron formation via quark coalescence, consistent with the formation of partonic flowing systems in these collisions.

Ultrarelativistic collisions of heavy ions at the Relativistic Heavy Ion Collider (RHIC) and the Large Hadron Collider (LHC) create the quark–gluon plasma (QGP), a short-lived state of strongly interacting partonic matter, thought to have existed a few microseconds after the Big Bang[1]. The interactions among partons in the QGP, combined with the initial spatial anisotropy of the overlap region of colliding ions, create anisotropic pressure gradients in the transverse plane of the collision. These anisotropic pressure gradients result in momentum anisotropy of the emitted particles[2]. The anisotropic particle emission is quantified using the Fourier decomposition of the azimuthal distribution of the final state particles[3]

$$\frac{\mathrm{d}N}{\mathrm{d}\varphi} \propto 1 + \sum_n 2v_n(p_\mathrm{T})\cos(n\varphi - n\Psi_n). \tag{1}$$

Here, $\varphi$ and $p_\mathrm{T}$ denote the azimuthal angle and transverse momentum of the emitted particles, respectively, while $\Psi_n$ is the azimuthal angle of the symmetry plane for the $n$-th harmonic. The largest contributions are the second and third Fourier coefficients, namely elliptic ($v_2$) and triangular ($v_3$) flow[4–6], which result from the elliptic and triangular shapes in the initial overlap region of the colliding nuclei. The anisotropic flow extends along pseudorapidity ($\eta \equiv -\ln(\tan\frac{\theta}{2})$, where $\theta$ is the polar angle of the particle), forming an elongated structure known as the ridge[7]. In heavy-ion collisions, precise measurements of $v_n$[4,6,8,9] and detailed comparisons with models employing relativistic viscous hydrodynamics reveal that the QGP behaves as a liquid with a viscosity to entropy density ratio close to the lowest theoretical value allowed[10,11].

In high-energy heavy-ion collisions, the $v_2$ coefficient of identified hadrons exhibits a characteristic mass dependence at low $p_\mathrm{T}$, meaning that more massive particles show lower $v_2$ values at a given $p_\mathrm{T}$[12–14]. Mass ordering arises from the interplay between average radial expansion velocity, anisotropic flow velocity, and thermal motion, which pushes heavier particles to higher $p_\mathrm{T}$[15,16]. This results in a mass-dependent reduction in $v_2$ at low $p_\mathrm{T}$ ($p_\mathrm{T} < 3.0$ GeV/$c$). In the intermediate $p_\mathrm{T}$ region ($3.0 < p_\mathrm{T} < 8.0$ GeV/$c$), a clear separation between the flow patterns of baryons (hadrons composed of three quarks or three antiquarks) and mesons (hadrons composed of quark–antiquark pairs) is observed

---

with $v_2^{\text{baryons}} > v_2^{\text{mesons}}$[14,17,18]. A physical process that can explain this distinctive grouping of hadron $v_2$ based on their valence quark number is hadron formation via quark coalescence[19,20]. In this process, a meson (baryon) is formed by combining two (three) quarks, and the meson (baryon) $v_2$ is obtained by combining the $v_2$ values of the two (three) quarks, as illustrated in Fig. 1. The experimental observation of baryon-meson grouping at intermediate $p_T$ is therefore interpreted as a consequence of a medium that includes a phase with collectively flowing partons.

Proton–proton and proton–nucleus collisions were used as a baseline to study the QGP in heavy-ion collisions at RHIC and the LHC, as QGP formation was not expected in small collision systems. However, striking similarities have been observed between numerous observables in both small collision systems and heavy-ion collisions at RHIC and LHC energies. These observables include the ridge[21–24], mass dependence of $v_2$ at low $p_T$[22,25,26], azimuthal angle correlations carried by multiple particles[27], and strangeness and baryon enhancement with increasing multiplicity[28]. These features are commonly considered indicators of QGP formation. The standard picture in heavy-ion collisions is that anisotropic flow is built up after the collision through final-state interactions among the partons combined with the initial spatial anisotropy of the overlap region of the colliding nuclei. In small collision systems, where the system evolution is shorter than in heavy-ion collisions and a QGP phase is not expected, a different scenario is proposed within the framework of Color Glass Condensate (CGC) effective theory[29]. According to this theory, the observed flow patterns come from the initial gluon momentum correlations in the colliding hadrons. These gluons scatter off specific regions, or color domains, during the collision and get a momentum boost in the same direction if they scatter from the same color domain. The current understanding is that initial-state momentum anisotropy alone cannot explain the existing data, and the measurements seem to favor the scenario of final-state effects driven by initial geometry[1,24], following a similar

scenario as in heavy-ion collisions. However, the impact of initial gluon momentum correlations on the development of anisotropic flow in small collision systems is not clear yet. At the same time, the precise mechanisms underlying the final-state effects remain unclear. The flow can develop during a partonic phase, transforming the initial spatial anisotropy into the measured flow[30–35], or it can originate via other mechanisms without the need for a deconfined phase, such as rescatterings among hadrons[36], via approaches involving initial state effects[37], or via different string dynamics implemented in the PYTHIA 8 event generator[38]. None of the measurements performed so far have been able to give a clear answer to this question.

This work takes a step further in resolving the puzzle of the origin of collective flow in small collision systems by investigating the possibility of a partonic phase and its role in the system's dynamic evolution. Utilizing the unique particle identification capabilities of the ALICE detector at the LHC[39,40], the elliptic flow ($v_2$) as a function of $p_T$ is presented for mesons ($\pi^\pm$, $K^\pm$, $K_S^0$) and baryons (p+$\overline{\text{p}}$, $\Lambda$ +$\overline{\Lambda}$) in pp and p–Pb collisions at a nucleon–nucleon center-of-mass energy ($\sqrt{s_{\text{NN}}}$) of 13 TeV (pp) and 5.02 TeV (p–Pb). In both pp and p–Pb systems, collisions are categorized into high-multiplicity (HM) and low-multiplicity (LM) events based on the charged particles detected within the pseudorapidity ranges $2.8 < \eta < 5.1$ and $-3.7 < \eta < -1.7$, respectively. Additionally, a selection criterion on the number of reconstructed and efficiency-corrected charged particles ($N_{\text{ch}}$) with $0.2 < p_T < 3.0$ GeV/$c$ at midrapidity ($|\eta| < 0.8$) is applied, resulting in the same average $N_{\text{ch}}$ ($\langle N_{\text{ch}} \rangle \approx 35$) in high-multiplicity events for both p-Pb and pp collisions. The event, track selection, and particle identification are discussed in the Methods section.

As $\Psi_n$ used in Eq. (1) cannot be experimentally determined, the $v_2$ of hadrons can be obtained using two-particle azimuthal angle correlations (2PC)[22] via the three-subevent method[41]. This method employs reference particles selected in the forward and backward rapidity regions in addition to the identified hadrons selected at midrapidity,

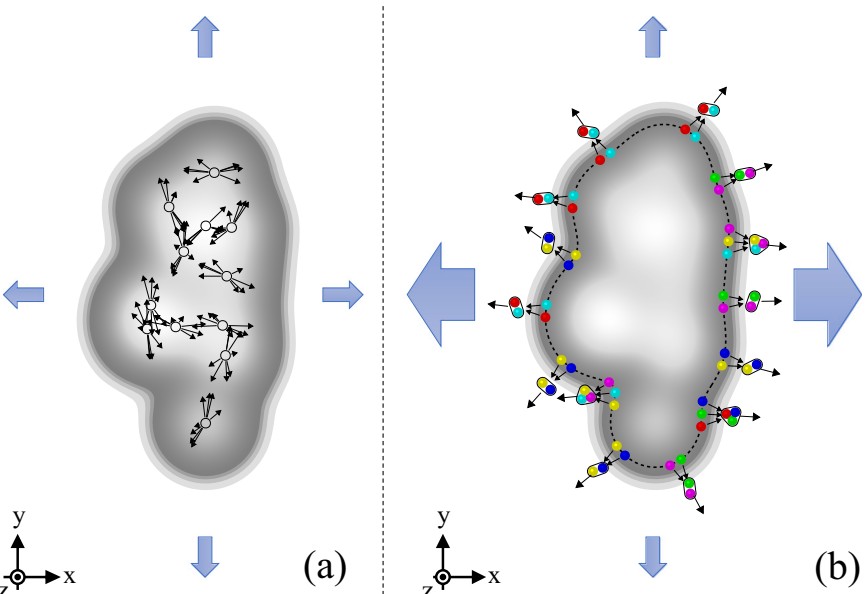

**Fig. 1 | Illustration of the overlap region**[67]**.** A schematic representation of the overlap region in a collision is shown in gray, along with overall particle emission patterns in the transverse (x-y) plane, represented by large arrows. **a** Non-flow sources: These are independent emissions, such as those from resonance decays or jets, where jets are collimated streams of hadrons created when a high-energy quark or gluon fragments after a collision. These effects lead to few-particle correlations but are not related to collective behavior in the system and have been subtracted from the final anisotropic flow measurements (see Correlation function and template fit method in the methods subsection for details). **b** Anisotropic flow: This illustrates the development of anisotropic flow in a partonic system, propagated to the level of hadrons via the quark coalescence process, which describes the experimental measurements in the intermediate $p_T$ range (~3-8 GeV/$c$). In this process, two or three flowing partons coalesce to form mesons or baryons, which then interact with each other. The large arrows represent the overall anisotropy of particle emission in the transverse plane, with stronger expansion along the short (x) axis.

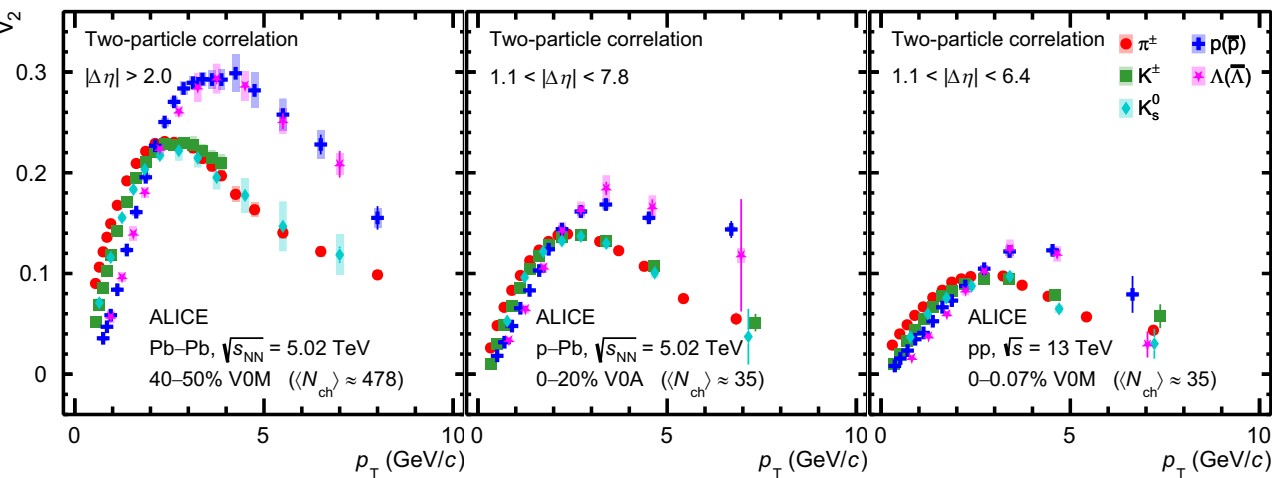

**Fig. 2 | $p_T$-differential $v_2$ measured with the two-particle correlation method[45] for mesons ($\pi^\pm$, $K^\pm$, $K_S^0$) and baryons (p+$\bar{p}$, $\Lambda$ +$\bar{\Lambda}$).** Left: results for semicentral Pb–Pb collisions at $\sqrt{s_{NN}}$ = 5.02 TeV. Middle: results in high-multiplicity p–Pb collisions at $\sqrt{s_{NN}}$ = 5.02 TeV. Right: Same for pp collisions at $\sqrt{s}$ = 13 TeV. $\langle N_{ch} \rangle$ is the average number of reconstructed, efficiency-corrected charged particles with $0.2 < p_T < 3.0$ GeV/$c$ at midrapidity ($|\eta| < 0.8$). Horizontal bars (boxes) represent the statistical (systematic) uncertainties.

allowing a significant pseudorapidity separation between the two correlated particles, $1.1 < |\Delta\eta| < 7.8$ in p–Pb, and $1.1 < |\Delta\eta| < 6.4$ in pp collisions. The difference in the maximum available $|\Delta\eta|$ separation between pp and p–Pb collisions is due to the different performance of the forward/backward detectors in different data-taking periods. However, for each collision system, the final results are estimated with varying $|\Delta\eta|$ separations and found to be consistent with each other. This large $\Delta\eta$ separation suppresses the non-flow contamination effects due to the azimuthal angle correlations between particles produced from resonance decays and jets[42]. To further minimize the remaining non-flow effects, a template fit method[23,43] is employed to fit the two-particle correlation functions. The non-flow template is obtained from the analysis of LM pp and p–Pb data and is explained in detail in section Methods. This method reduces the non-flow contribution to the final $v_2$ results for identified particles to 6% for $p_T < 0.6$ GeV/$c$ and to less than 1% at higher $p_T$. These estimates are based on applying the template-fit method to a pure non-flow model using the PYTHIA 8 event generator[44].

## Results

Figure 2 presents the $p_T$ -differential $v_2$ measurement for mesons ($\pi^\pm$, $K^\pm$, $K_S^0$) and baryons (p+$\bar{p}$, $\Lambda$ +$\bar{\Lambda}$) in semicentral Pb–Pb[45], HM p–Pb and pp collisions. For Pb–Pb measurements, the two-particle correlation with $|\Delta\eta| > 2.0$ separation is used[45], whereas for p–Pb and pp collisions, a $|\Delta\eta| > 1.1$ separation is applied. It was tested that varying the $|\Delta\eta|$ separation does not significantly alter the results in Pb–Pb collisions[45]. Figure 2 shows a clear similarity in the characteristic features of $v_2$ among the three collision systems. The difference in the magnitude of $v_2$ among the three collision systems is consistent with previous measurements at the LHC[27]. For the low $p_T$ region, $p_T < 2.0$ GeV/$c$, a clear mass ordering of the $v_2$ coefficients is observed, providing significant evidence of radial flow in small collision systems. The presence of radial flow in small collision systems is also supported by particle spectra measurements[46]. Around $2.0 < p_T < 3.0$ GeV/$c$, the $v_2$ coefficients of different particle species begin to cross. Beyond $p_T > 2.5$ GeV/$c$, the $v_2$ coefficients of baryons (p+$\bar{p}$, $\Lambda$ +$\bar{\Lambda}$) are consistent with each other within 1 standard deviation ( ~ 1$\sigma$) up to 10 GeV/$c$ in Pb–Pb and p–Pb collisions and up to 6 GeV/$c$ in pp collisions. At the same time, the $v_2$ of mesons ($\pi^\pm$, $K^\pm$, $K_S^0$) are compatible within ~ 1$\sigma$ at $p_T > 2(3)$ GeV/$c$ for Pb–Pb and p–Pb (pp) collisions. Moreover, the $v_2$ of baryons is

larger than that of mesons by ~5$\sigma$ at intermediate and higher $p_T$ ($p_T > 3.0$ GeV/$c$) in all three collision systems. In heavy-ion collisions, such distinctive baryon-meson $v_2$ grouping at intermediate $p_T$ is explained by anisotropic flow development at the quark level, followed by particle production via the quark-coalescence mechanism[14,17,18].

Existing measurements with identified particles in small collision systems presented either a single baryon or meson $v_2$, limiting the opportunity to explore potential groupings among baryons and mesons[47–49]. Other similar measurements have not shown a clear grouping and splitting of baryon and meson $v_2$ at intermediate $p_T$ in small systems, differing from similar measurements in heavy-ion collisions[25]. This difference may arise from difficulties in accounting for non-flow effects in small systems. For example, previous measurements either applied no non-flow removal technique[26,49], or were based on the subtraction of the low-multiplicity correlation functions from high-multiplicity ones[22,25,47], under the assumption that the former include only non-flow effects and no significant long-range correlations. However, recent ALICE measurements show that long-range correlations persist even in pp and p–Pb collisions with very low multiplicity ($\langle N_{ch} \rangle \approx 10$)[50]. This implies that the subtraction method also removes, along with non-flow effects, part of the real correlation signal, thus leading to over-subtraction. Such over-subtraction can vary by particle type and can potentially create a particle-type-dependent $v_2$ pattern that does not originate from a true physics effect. As a result, subtraction-based methods are unreliable for studying the baryon−meson $v_2$ splitting in small collision systems. The results presented in Fig. 2, after removal of non-flow effects (see Correlation function and template fit method in the methods subsection for details), show a distinctive baryon−meson $v_2$ grouping (within 1$\sigma$) and a significant splitting (~5$\sigma$) at intermediate $p_T$ in both p–Pb and pp collisions at the LHC, similar to what is observed in heavy-ion collisions.

In refs. 25,26, the number-of-constituent-quark (NCQ) scaling of $v_2$ has also been studied. This scaling was initially attributed to hadron production via the coalescence of thermal partons in heavy-ion collisions[19,20,51]. However, advanced coalescence models incorporate the recombination of thermal quarks with shower quarks originating from jet-medium interactions to describe the $p_T$ spectra and $v_2$ of identified particles over a broad $p_T$ range[52], differing from the coalescence mechanism[19,20,51] associated with NCQ scaling of $v_2$. In addition,

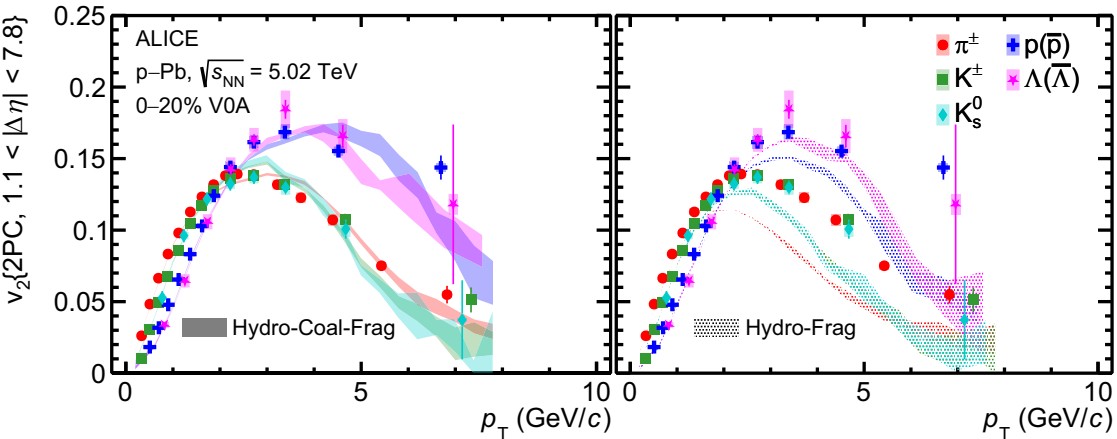

**Fig. 3 | $v_2$ in high-multiplicity p–Pb collisions.** $p_T$-differential $v_2$ measured with two-particle correlation for mesons ($\pi^\pm$, $K^\pm$, $K_s^0$) and baryons (p+$\bar{p}$, $\Lambda$ +$\bar{\Lambda}$) in high-multiplicity p–Pb collisions at $\sqrt{s_{NN}}$ = 5.02 TeV. Horizontal bars (boxes) represent the statistical (systematic) uncertainties. Comparisons with the calculations from the Hydro-Coal-Frag model (left) and the Hydro-Frag model (right) are also presented[54,55]. Only statistical uncertainties are shown for the calculations.

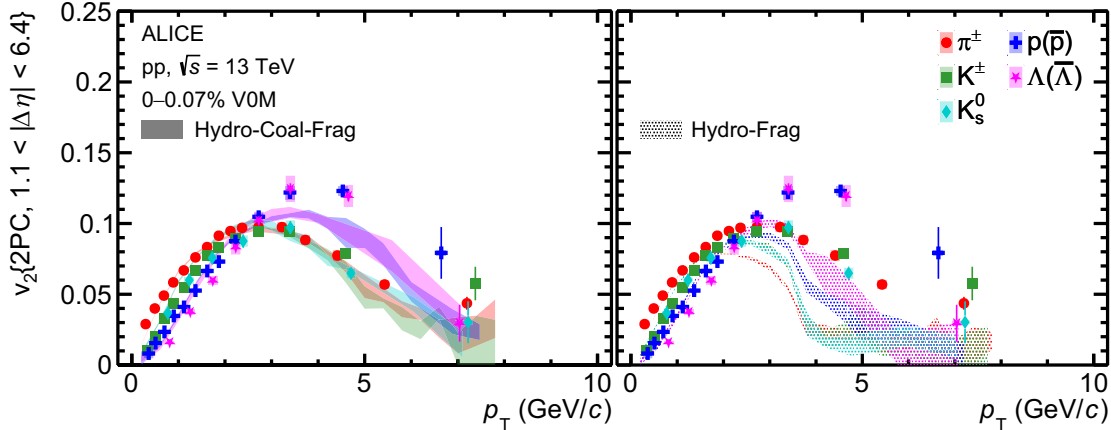

**Fig. 4 | $v_2$ in high-multiplicity pp collisions.** $p_T$-differential $v_2$ measured with two-particle correlation for mesons ($\pi^\pm$, $K^\pm$, $K_s^0$) and baryons (p+$\bar{p}$, $\Lambda$ +$\bar{\Lambda}$) in high-multiplicity pp collisions at $\sqrt{s}$ = 13 TeV. Horizontal bars (boxes) represent the statistical (systematic) uncertainties. Comparisons with the calculations from the Hydro-Coal-Frag model (left) and the Hydro-Frag model (right) are also presented[54,55]. Only statistical uncertainties are shown for the calculations.

contributions from radial flow and jet fragmentation at intermediate $p_T$ can also lead to deviations from NCQ scaling. Notably, the ALICE measurements exhibit deviations from NCQ scaling at the level of ± 20% in Pb–Pb collisions[14,53]. This underscores the need for a better understanding of this scaling as evidence of partonic collectivity in relativistic collisions.

In Figs. 3 and 4, the $v_2$ measurements are compared with the state-of-the-art calculations using the Hydro-Coal-Frag hybrid model[54,55] for p–Pb and pp collisions, respectively. At low $p_T$, this model incorporates the hydrodynamic evolution of a quark–gluon plasma with a partonic equation of state, followed by the formation of quarks before hadronization. At high $p_T$, it accounts for interactions between high-energy partons and the medium using the linear Boltzmann transport (LBT) model, combined with hadronization via quark fragmentation. The intermediate $p_T$ hadrons are produced from the coalescence of quarks originating from both hydrodynamic evolution and jet-medium interactions. Finally, hadronic interactions occur after hadronization. A more detailed description of this model can be found in the methods subsection. This model provides a comprehensive explanation for both hadron production and anisotropic flow over a wide $p_T$ range in high-energy heavy-ion collisions[52]. It emphasizes the crucial role of

partonic flow and particle production through quark coalescence in heavy-ion collisions, where the QGP is formed. In Fig. 3, the model parameters are tuned to describe the $p_T$ spectra of identified hadrons in high-multiplicity p–Pb collisions at $\sqrt{s_{NN}}$ = 5.02 TeV[54]. The figure demonstrates that the Hydro-Coal-Frag model successfully reproduces the baryon-meson $v_2$ splitting and grouping features for $p_T$ < 8 GeV/$c$ as observed in the experimental data. In contrast, the calculation from the Hydro-Frag model, which does not include the quark-coalescence process, strongly underestimates the $v_2$ coefficients of all identified hadrons for $p_T$ > 4 GeV/$c$. Moreover, despite parameter adjustments, the Hydro-Frag model fails to even qualitatively reproduce the baryon-meson $v_2$ splitting and grouping at intermediate $p_T$[55].

The comparison between the measurements and the model prediction for pp collisions at $\sqrt{s}$ = 13 TeV is presented in Fig. 4. The model parameters are calibrated using $p_T$ spectra of identified hadrons from a different multiplicity interval than the one used in this paper. Still, the Hydro-Coal-Frag picture can explain the mass ordering of $v_2$ for $p_T$ up to 3 GeV/$c$ combined with the crossing between $v_2$ of baryons and mesons at $p_T \approx 3$ GeV/$c$, consistent with the data. Most importantly, the baryon-meson splitting and grouping of $v_2$ can be qualitatively reproduced by the Hydro-Coal-Frag model up to approximately 6–7

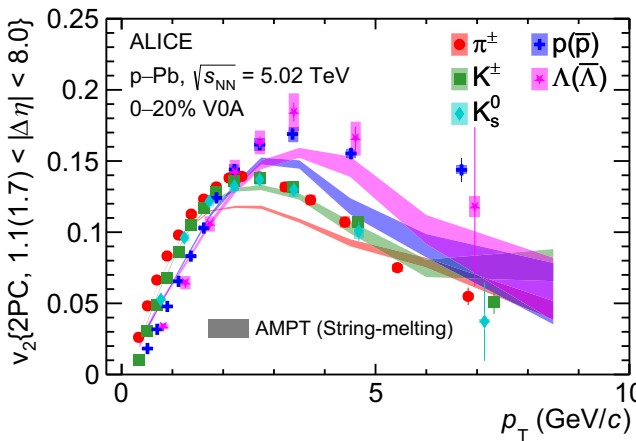

**Fig. 5 | $v_2$ in high-multiplicity p–Pb collisions compared with AMPT calculations.** $p_T$-differential $v_2$ measured with two-particle correlation for mesons ($\pi^\pm$, $K^\pm$, $K^0_S$) and baryons (p+$\bar{p}$, $\Lambda$ +$\bar{\Lambda}$) in high-multiplicity p–Pb collisions at $\sqrt{s_{NN}}$ = 5.02 TeV. Horizontal bars (boxes) represent the statistical (systematic) uncertainties. Comparison with the calculations from the AMPT String-melting model[58] is also presented. The AMPT curves are obtained by applying the same template fit method to the correlation distributions as used in the data analysis[58]. Only statistical uncertainties are shown for the AMPT calculations.

GeV/$c$. In contrast, neither of these features is captured by the Hydro-Frag model. The crossing between $v_2$ of different particles occurs at about 2 GeV/$c$, and only mass ordering is observed in this calculation[55]. The ordering is reversed for 2.5 < $p_T$ < 5 GeV/$c$ compared to $p_T$ < 2.5 GeV/$c$, similar to the Hydro-Frag calculations in p–Pb collisions shown in Fig. 3. Therefore, these results provide evidence of hadronization via coalescence of hydrodynamically flowing quarks in small collision systems at the LHC.

Other theoretical calculations that predict a flow pattern in high multiplicity events of small collision systems have also been studied. Since the presented measurements are based on long-range (i.e., large |$\Delta\eta$| separation) two-particle correlations, they are predominantly influenced by geometry-driven effects, as the initial momentum anisotropy in the CGC approach generates only short-range correlations[37]. The hadronic rescatterings in UrQMD can mimic the long-range two-particle correlation and mass dependence of $v_2$ at low $p_T$ [36], but do not generate any baryon-meson $v_2$ splitting and grouping. The PYTHIA 8 model, with color ropes enabled in the hadronization process, can describe the strangeness enhancement in pp collisions[28] without the formation of a QGP[56], but cannot generate long-range correlations. Interestingly, the string-string repulsion in the string-shoving version of PYTHIA 8 can generate long-range two-particle correlations[38,57], but it produces negative flow coefficients after the template fit, in contrast to the positive flow coefficients observed in the data. In small systems, transport models like AMPT[35], which generate only a few partonic interactions during system evolution and incorporate the quark-coalescence model of hadronization, can approximately describe the $v_2$ at low $p_T$. However, they fail to even qualitatively explain the baryon-meson $v_2$ grouping and splitting feature observed at intermediate $p_T$ as shown in Fig. 5[58]. This indicates that anisotropic flow is developed in a dense partonic system and propagated to the level of hadrons via the quark coalescence process.

In summary, the $v_2$ for the identified hadrons in high-multiplicity p–Pb collisions at $\sqrt{s_{NN}}$ = 5.02 TeV and pp collisions at $\sqrt{s}$ = 13 TeV has been presented as a function of $p_T$ and compared with measurements in semicentral Pb–Pb collisions at $\sqrt{s_{NN}}$ = 5.02 TeV. A characteristic grouping (within ~1$\sigma$) and splitting (with ~5$\sigma$) of $v_2$ for mesons ($\pi^\pm$, $K^\pm$, $K^0_S$) and baryons (p+$\bar{p}$, $\Lambda$ +$\bar{\Lambda}$) at intermediate $p_T$, similar to

measurements in heavy-ion collisions, is observed in both p–Pb and pp collisions. The Hydro-Coal-Frag model, incorporating partonic flow and quark coalescence, provides the best possible description of the data to date in both heavy-ion and small collision systems. Alternative approaches fail to even qualitatively reproduce the baryon-meson $v_2$ splitting and grouping at intermediate $p_T$. The presented measurements and the comparisons with available theoretical model calculations provide evidence that the system created in high-multiplicity p–Pb and pp collisions includes a stage with hydrodynamically flowing partons, similar to the one observed in heavy-ion collisions.

## Methods

### Event selection

The analyzed data samples are from pp collisions at $\sqrt{s}$ = 13 TeV and p–Pb collisions at $\sqrt{s_{NN}}$ = 5.02 TeV, collected by the ALICE detector during the LHC Run 2 data-taking campaign between 2016 and 2018. An extensive description of all subdetectors of ALICE can be found in refs. 39,40. The collected data are classified based on the specific triggering conditions. Minimum bias (MB) events for both pp and p–Pb collisions are triggered using a coincidence signal in the two scintillator arrays of the V0 detector, which cover the pseudorapidity ranges 2.8 < $\eta$ < 5.1 (V0A) and −3.7 < $\eta$ < −1.7 (V0C), respectively. To avoid the possibility of overlap between HM and LM event classes, additional requirements on the number of reconstructed and efficiency-corrected charged particles ($N_{ch}$) within the acceptance of |$\eta$| < 0.8 and transverse momentum 0.2 < $p_T$ < 3.0 GeV/$c$ are introduced. For p–Pb collisions, the collisions of the 0–20% and 60–100% V0A-centrality are used as the HM and LM events, with an additional criterion of $N_{ch}$ < 20 applied to the LM sample. In pp collisions, HM events are selected from the 0.07% of events with the highest multiplicity. This selection uses a special HM trigger based on the amplitude of the V0M detector arrays (V0A + V0C) and requires $N_{ch}$ > 25. For the LM event class, MB events with $N_{ch}$ < 20 are selected. Only events with a reconstructed primary vertex (PV) within ± 10 cm from the nominal interaction point along the beam line are selected. Background events due to interaction between the beam and the residual gas molecules in the beam pipe are removed using the information from the Silicon Pixel Detector (SPD) and V0 detectors. The out-of-bunch pileup events are rejected using a correlation between the multiplicities in the V0 and Forward-Multiplicity-Detector (FMD)[39]. The in-bunch pileup is reduced by rejecting events with multiple vertices. The event selections result in a sample of 226 × 10^6 HM and 572 × 10^6 LM pp collisions, corresponding to integrated luminosities of approximately 4 nb$^{-1}$ and 10 nb$^{-1}$, respectively[59]. For p–Pb, the analyzed data sample consists of 101 × 10^6 HM collisions and 198 × 10^6 LM collisions, corresponding to integrated luminosities of about 0.05 nb$^{-1}$ and 0.09 nb$^{-1}$, respectively[60]. In this analysis, the HM events are used for the flow measurements, whereas the LM events are used for the baseline nonflow estimation.

### Track reconstruction

The charged particles at midrapidity are reconstructed using the Inner Tracking System (ITS) and the Time Projection Chamber (TPC). The selected tracks have at least 70 TPC space points (out of a maximum of 159) fitted for track reconstruction with $\chi^2$ per degree of freedom lower than four. The reconstructed tracks from the TPC and the ITS must coincide to ensure the consistency of the reconstruction method. Additionally, a minimum of two hits are required in the ITS to improve the momentum resolution. The pseudorapidity of the selected tracks is required to be within |$\eta$| < 0.8 to reject tracks with reduced reconstruction efficiency at the detector edges. To reduce the contamination from secondary particles, the distance-of-closest approach (DCA) of the selected tracks to the PV must be within 2 cm in the longitudinal direction. Furthermore, a $p_T$-dependent DCA selection in the transverse plane, ranging from 0.2 cm at $p_T$ = 0.2 GeV/$c$ to 0.02 cm at

$p_T$ = 5.0 GeV/$c$, is applied. This criterion suppresses the residual contamination from secondary particles from weak decays and interactions in the detector material. The reference particles used for the construction of long-range di-hadron correlation functions are selected from the FMD detector segments located at forward (FMD1,2 with 1.7 < $\eta$ < 5.1) and backward (FMD3 with −3.1 < $\eta$ < −1.7) rapidity regions.

## Particle identification and reconstruction

The charged tracks are identified as $\pi^\pm$, $K^\pm$, and p+$\overline{\text{p}}$ based on the specific energy loss (d$E$/d$x$) information in the TPC and the velocity information from the Time-of-Flight (TOF) detector. A Bayesian approach[45] is used to identify particle species at a given $p_T$ using the correlation of the normalized differences between the measured and the expected signal in the TPC ($n\sigma_{TPC}$) and the TOF ($n\sigma_{TOF}$), respectively. In this method, the signals are converted into probabilities and folded with the expected abundances (priors) of each particle species. To ensure high purity of the selected sample, a minimal probability threshold of 0.95 for $\pi^\pm$ and 0.85 for $K^\pm$ and p+$\overline{\text{p}}$ is set. In addition, the tracks with proper TOF information are required to be within $|n\sigma_{TPC}|$ < 3 and $|n\sigma_{TOF}|$ < 3. The resulting purity, estimated using Monte Carlo (MC) simulations, is higher than 95% for $\pi^\pm$ for 0.2 < $p_T$ < 10 GeV/$c$, above 80% for $K^\pm$ for 0.3 < $p_T$ < 10 GeV/$c$, and reaches values larger than 90% for p+$\overline{\text{p}}$ for 0.5 < $p_T$ < 10 GeV/$c$. The high purity of the studied sample reduces the uncertainties due to particle misidentification. The $K_S^0$ and $\Lambda$ +$\overline{\Lambda}$ are weakly decaying neutral particles, reconstructed by calculating the invariant mass of the daughter particles from the most probable decay channels of $K_S^0 \rightarrow \pi^+ + \pi^-$ and $\Lambda \rightarrow$ p + $\pi^-$ ($\overline{\Lambda} \rightarrow \overline{\text{p}} + \pi^+$) with branching ratios of 69.2% (±0.05%) and 64.1% (±0.5%)[61], respectively. The combinatorial background is suppressed by using a set of selection criteria on the decay topology used in the previous $K_S^0$ and $\Lambda$ measurements in ALICE[28]. The $K_S^0$ and $\Lambda$ +$\overline{\Lambda}$ candidates are selected within the rapidity range $|y|$ < 0.5 inside the TPC, and the daughter tracks are used to reconstruct the secondary decay vertex (SV) in the offline reconstruction. The SV is required to be more than 0.5 cm away from the PV, and the reconstructed proper lifetime, defined as $mL/p$, ($m$ being the particle mass, $L$ the distance between the primary and secondary vertices, and $p$ the particle momentum) should be smaller than 20 cm and 30 cm for $K_S^0$ and $\Lambda$ ($\overline{\Lambda}$) candidates, respectively. The oppositely charged daughter tracks are combined only if they are identified as pions or protons based on the TPC d$E$/d$x$ hypothesis (<3$\sigma$). A set of topological cuts, such as the distance of closest approach (DCA) of the daughter tracks to the primary vertex (> 0.06 cm), DCA between the daughter tracks (<1 cm), and cosine of the pointing angle, which is the angle between the momentum direction of the mother particle and the direction from the PV to the decay point (>0.97 for $K_S^0$ and >0.995 for $\Lambda$ ($\overline{\Lambda}$)) are applied to reduce the combinatorial background contribution to the invariant mass spectrum.

## Correlation function and template fit method

The correlation function is obtained between two sets of particles classified as trigger and associated. Trigger particles are used as a reference, and the angular distribution of associated particles is measured relative to the trigger particles[22]. In this analysis, the 2D correlation function is constructed as a function of the difference in azimuthal angle $\Delta\varphi = \varphi_{trigger} - \varphi_{associated}$ and pseudorapidity $\Delta\eta = \eta_{trigger} - \eta_{associated}$ with trigger and associated particles from different detectors. Three sets of correlation functions are constructed to estimate the $v_2$ of identified particles ($\pi^\pm$, $K^\pm$, p+$\overline{\text{p}}$, $K_S^0$, and $\Lambda$ +$\overline{\Lambda}$). The identified particles in the TPC are correlated with unidentified reference particles reconstructed within the FMD acceptance in positive (FMD1,2) and negative (FMD3) rapidity regions to construct two sets (TPC–FMD1,2 and TPC–FMD3) of correlation functions. The third correlation function is constructed using two reference particles from the FMD1,2 and FMD3

detector segments. The pair acceptance effect due to the finite size of the detectors is corrected by dividing the same-event correlation functions with mixed-event correlation functions. The mixed-event correlation function is constructed by correlating the trigger particles in one event with the associated particles from other events belonging to the same multiplicity event class and with PV within a given 2 cm wide interval. The mixed event is normalized using a constant estimated by averaging over all $\Delta\varphi$ bins at the $\Delta\eta$ value where the mixed event correlation function reaches its maximum. The corrected correlation function is obtained as a ratio of the same and mixed event correlation functions for each PV position. The final correlation function for an event class (HM or LM events) is calculated after averaging the correlation functions over all PV positions. For each of the 2D correlation functions (TPC–FMD1,2, TPC–FMD3, and FMD1,2–FMD3), the projection along the $\Delta\varphi$ axis is calculated for both HM and LM cases. The $\Delta\varphi$ projections from LM collisions serve as a template for subsequent fitting of the $\Delta\varphi$ projections from HM collisions to extract the $v_2$ coefficients using the template fit method. The template fit method assumes that high-multiplicity (HM) collisions are a superposition of low-multiplicity (LM) collisions, which primarily contain non-flow effects with some residual flow, along with an additional flow modulation, i.e.,

$$Y(\Delta\varphi)^{HM} = FY(\Delta\varphi)^{LM} + G\left[1 + \sum_{n=2}^{\infty} 2V_{n\Delta} \cos(n\Delta\varphi)\right], \qquad (2)$$

where Y($\Delta\varphi$)$^{HM}$ and Y($\Delta\varphi$)$^{LM}$ are the one dimensional $\Delta\varphi$ projections of the 2D correlation functions obtained in HM and LM collisions with $F$ and $G$ being the scaling factors. The $V_{n\Delta}$ coefficients are estimated by fitting the correlation function with the equation (2). The scaling factors $F$ and $G$ are free parameters in this template fit procedure. The final $v_2$ of the identified particles in the TPC is calculated by combining the $V_{2\Delta}$ estimated from the TPC–FMD1,2, TPC–FMD3, and FMD1,2–FMD3 correlation functions

$$v_2^{PID}(p_T) = \sqrt{\frac{V_{2\Delta}^{TPC-FMD1,2} V_{2\Delta}^{TPC-FMD3}}{V_{2\Delta}^{FMD1,2-FMD3}}}. \qquad (3)$$

In this work, all available non-flow suppression methods (low-multiplicity subtraction[22,25,47], template fit[23], and improved template fit[62]) have been tested, and the residual non flow has been estimated using PYTHIA8 for each method. Among these, the template fit provides the most effective non flow subtraction, yielding the lowest residual non flow (-5-7%) across the considered kinematic range. This residual non flow has been included in the systematic uncertainties. The inclusion of the remaining non flow enables better comparisons with theoretical models and supports robust, data driven physics conclusions.

## Systematic uncertainty

The systematic uncertainties are evaluated by varying the event, track, and PID selection criteria with respect to the default ones, one at a time. For each variation, the difference between the default and varied result is estimated using the Barlow criterion[63], and a difference higher than 1$\sigma$ is considered as a possible source of systematic uncertainty in the measurement. The Barlow difference is calculated for each particle species and for each $p_T$ interval for which the final $v_2$\{2PC\} results are presented in this paper. The Barlow ratio is calculated as

$$B = \frac{|v_2^{default} - v_2^{syst}|}{\sqrt{|\sigma_{default}^2 - \sigma_{syst}^2|}}. \qquad (4)$$

**Table 1 | Systematic uncertainties in p–Pb collisions**

| | $v_2\{2PC, 1.1 < |\Delta\eta| < 7.8\}$ | | | | |
|---|---|---|---|---|---|
| Uncertainty source | $\pi^{\pm}$ | $K^{\pm}$ | $p+\bar{p}$ | $K_S^0$ | $\Lambda + \bar{\Lambda}$ |
| Primary vertex position | 1–2% | 1–3% | 1–2% | negl. | negl. |
| FMD–V0 correlation | 0–4% | 0–2% | 0–2% | 1–4% | 2–4% |
| Primary track quality | negl. | negl. | negl. | – | – |
| Bayesian threshold | negl. | negl. | negl. | – | – |
| Topological criteria | – | – | – | 1–2% | 1–4% |
| Invariant mass acceptance | – | – | – | negl. | 3–4% |
| Invariant mass fit | – | – | – | negl. | 1–3% |
| Template variation | negl. | negl. | 1–2% | negl. | negl. |
| Secondary contamination | 1% | 1% | 1% | 1% | 1% |
| Total | 2–4% | 2–3% | 1–3% | 2–5% | 5–7% |

The minimum and maximum values of the relative systematic uncertainties from individual sources for $\pi^{\pm}$, $K^{\pm}$, $p+\bar{p}$, $K_S^0$, and $\Lambda + \bar{\Lambda}$ in p–Pb collisions. Percentage ranges are given to account for variations with $p_T$. The fields marked as "negl." (negligible) denotes that the uncertainties have been tested but are not statistically significant.

**Table 2 | Systematic uncertainties in pp collisions**

| | $v_2\{2PC, 1.1 < |\Delta\eta| < 6.4\}$ | | | | |
|---|---|---|---|---|---|
| Uncertainty source | $\pi^{\pm}$ | $K^{\pm}$ | $p+\bar{p}$ | $K_S^0$ | $\Lambda + \bar{\Lambda}$ |
| Primary vertex position | 0–1% | negl. | negl. | 2–3% | 1–3% |
| FMD–V0 correlation | 1–2% | negl. | 0–2% | 1–4% | 1–3% |
| Primary track quality | 0–2% | negl. | negl. | – | – |
| Bayesian threshold | negl. | 0–2% | negl. | – | – |
| Topological criteria | – | – | – | 2–4% | 1–5% |
| Invariant mass acceptance | – | – | – | negl. | 1–3% |
| Invariant mass fit | – | – | – | 1–2% | 3–4% |
| Template variation | 0–4% | 0–2% | 1–3% | 1–3% | 2–4% |
| Secondary contamination | 1% | 1% | 1% | 1% | 1% |
| Total | 2–5% | 2% | 2–4% | 4–6% | 4–8% |

The minimum and maximum values of the relative systematic uncertainties from individual sources for $\pi^{\pm}$, $K^{\pm}$, $p+\bar{p}$, $K_S^0$, and $\Lambda + \bar{\Lambda}$ in pp collisions. Percentage ranges are given to account for variations with $p_T$. The fields marked as "negl." (negligible) denotes that the uncertainties have been tested but are not statistically significant.

If the Barlow difference is higher than $1\sigma$ for more than 1/3 of the total $p_T$ intervals for any species, the contribution of that particular systematic source is included in the uncertainty of the final result. Otherwise, the contribution from that systematic source is considered negligible and does not contribute to the final systematic uncertainty. The minimum and maximum values of the relative systematic uncertainties from individual sources are presented in Tables 1 and 2 for p–Pb and pp collisions, respectively. The systematic sources listed in the tables from top to bottom include different PV intervals used for event selection, correlation between multiplicity from V0 and FMD detectors to reduce contamination in the FMD, track selection criteria, and particle identification criteria affecting the purity of the $\pi^{\pm}$, $K^{\pm}$ and $p+\bar{p}$ samples. Other factors are topological reconstruction criteria, invariant mass reconstruction and fitting requirements impacting the signal-to-background ratios for $K_S^0$ and $\Lambda + \bar{\Lambda}$ candidates, the definition of the low-multiplicity template used for non-flow removal, and the

estimation of residual secondary contamination in the FMD. The latter is done using a Monte Carlo event generator, by transporting the generated particles through GEANT3-simulated detector response and performing track reconstruction in the ALICE framework. The contributions from the different sources are added in quadrature to estimate the total systematic uncertainty.

**Hydro+Coal+Frag model description**

The Hydro-Coal-Frag model[52,54,55] provides a unified theoretical framework for hadron production in high-energy nuclear collisions, bridging soft and hard processes across transverse momentum ($p_T$) regimes. It describes low-$p_T$ hadrons via viscous hydrodynamics, intermediate-$p_T$ hadrons through quark coalescence, and high-$p_T$ hadrons via string fragmentation. The modeling sequence begins with the TRENTo model[64], which generates event-by-event initial entropy profiles based on the nuclear geometry. These profiles serve as initial conditions for the (2+1)-dimensional viscous hydrodynamics model VISH2+1[65], which governs the space-time evolution of the quark-gluon plasma (QGP). As the system cools toward the hydrodynamic freeze-out temperature, thermal hadrons are emitted according to the Cooper-Frye prescription[65], and thermal partons are sampled at low transverse momentum ($p_T$) for subsequent hadronization. High-$p_T$ partons (hard partons), generated using PYTHIA8, traverse the quark-gluon plasma and undergo medium-induced interactions, which are modeled using the Linear Boltzmann Transport (LBT) framework[66].

For intermediate-$p_T$ hadrons, the quark coalescence mechanism is used to recombine thermal–thermal, thermal–hard, and hard–hard partons produced by the hydrodynamics and LBT processes. Meson and baryon momentum distributions are derived from Wigner functions, which encode the spatial and momentum proximity of coalescing partons. Excited hadronic states, formed according to the invariant masses of parton pairs, subsequently decay into ground states, with binding energy differences and conservation of energy and momentum explicitly taken into account. Remaining hard partons without coalescence partners generate high-$p_T$ hadrons via string fragmentation. The transverse momentum ($p_T$) cut-off values for thermal parton sampling at freeze-out, along with the criteria governing whether partons undergo coalescence or fragmentation following the LBT stage, and the gluon virtuality parameters, are tuned to reproduce the $p_T$ spectra of pions, kaons, and protons, as well as the $(p(\bar{p})/\pi^{\pm})$ ratio in the intermediate-$p_T$ region of high-multiplicity p–p and p–Pb collisions at the LHC. The final hadronic evolution, including scatterings and resonance decays, is simulated using the Ultrarelativistic Quantum Molecular Dynamics (UrQMD) model.

**AMPT calculations**

Figure 5 presents the $p_T$-differential $v_2$ measured from two-particle correlations for mesons ($\pi^{\pm}$, $K^{\pm}$) and baryons ($p+\bar{p}$, $\Lambda + \bar{\Lambda}$) in HM p–Pb collisions at $\sqrt{s_{NN}} = 5.02$ TeV, compared with estimations from the AMPT string melting model[58]. The AMPT curves are obtained by applying the same template fit method to the correlation distributions as used in the analysis of the data. Both the data and AMPT calculations[58] select particles within similar rapidity regions, allowing a pseudorapidity separation between the two correlated particles of $1.1 < |\Delta\eta| < 7.8$ in the data and $1.7 < |\Delta\eta| < 8.0$ in the AMPT simulations.

## Data availability

This manuscript has associated data in a HEPData repository at: https://www.hepdata.net/record/ins2848254.

## Code availability

This manuscript has associated code/software in a data repository. The code/software used for the analysis is publicly available on the github

repository, at the links https://github.com/alisw/AliRoot and https://github.com/alisw/AliPhysics.

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

## Acknowledgements

The ALICE Collaboration would like to thank Huichao Song and Wenbin Zhao for providing the latest calculations from the state-of-the-art models. The ALICE Collaboration would like to thank all its engineers and technicians for their invaluable contributions to the construction of the experiment and the CERN accelerator teams for the outstanding performance of the LHC complex. The ALICE Collaboration gratefully acknowledges the resources and support provided by all Grid centers and the Worldwide LHC Computing Grid (WLCG) collaboration. The ALICE Collaboration acknowledges the following funding agencies for their support in building and running the ALICE detector: A.I. Alikhanyan National Science Laboratory (Yerevan Physics Institute) Foundation (ANSL), State Committee of Science and World Federation of Scientists (WFS), Armenia; Austrian Academy of Sciences, Austrian Science Fund (FWF): [M 2467-N36] and Nationalstiftung für Forschung, Technologie und Entwicklung, Austria; Ministry of Communications and High Technologies, National Nuclear Research Center, Azerbaijan; Conselho Nacional de Desenvolvimento Científico e Tecnológico (CNPq), Financiadora de Estudos e Projetos (Finep), Fundação de Amparo à Pesquisa do Estado de São Paulo (FAPESP) and Universidade Federal do Rio Grande do Sul (UFRGS), Brazil; Bulgarian Ministry of Education and Science, within the National Roadmap for Research Infrastructures 2020–2027 (object CERN), Bulgaria; Ministry of Education of China (MOEC), Ministry of Science & Technology of China (MSTC) and National Natural Science Foundation of China (NSFC), China; Ministry of Science and Education and Croatian Science Foundation, Croatia; Centro de Aplicaciones Tecnológicas y Desarrollo Nuclear (CEADEN), Cubaenergía, Cuba; Ministry of Education, Youth and Sports of the Czech Republic, Czech Republic; The Danish Council for Independent Research | Natural Sciences, the VILLUM FONDEN and Danish National Research Foundation (DNRF), Denmark; Helsinki Institute of Physics (HIP), Finland; Commissariat à l'Energie Atomique (CEA) and Institut National de Physique Nucléaire et de Physique des Particules (IN2P3) and Center National de la Recherche Scientifique (CNRS), France; Bundesministerium für Forschung, Technologie und Raumfahrt (BMFTR) and GSI Helmholtzzentrum für Schwerionenforschung GmbH, Germany; General Secretariat for Research and Technology, Ministry of Education, Research and Religions, Greece; National Research, Development and Innovation Office, Hungary; Department of Atomic Energy Government of India (DAE), Department of Science and Technology, Government of India (DST), University Grants Commission, Government of India (UGC) and Council of Scientific and Industrial Research (CSIR), India; National Research and Innovation Agency - BRIN, Indonesia; Istituto Nazionale di Fisica Nucleare (INFN), Italy; Japanese Ministry of Education, Culture, Sports, Science and Technology (MEXT) and Japan Society for the Promotion of Science (JSPS) KAKENHI, Japan; Consejo Nacional de Ciencia (CONACYT) y Tecnología, through Fondo de Cooperación Internacional en Ciencia y Tecnología (FONCICYT) and Dirección General de Asuntos del Personal Academico (DGAPA), Mexico; Nederlandse Organisatie voor Wetenschappelijk Onderzoek (NWO), Netherlands; The Research Council of Norway, Norway; Pontificia Universidad Católica del Perú, Peru; Ministry of Science and Higher Education, National Science Center and WUT ID-UB, Poland; Korea Institute of Science and Technology Information and National Research Foundation of Korea (NRF), Republic of Korea; Ministry of Education and Scientific Research, Institute of Atomic Physics, Ministry of Research and Innovation and Institute of Atomic Physics and Universitatea Nationala de Stiinta si Tehnologie Politehnica Bucuresti, Romania; Ministry of Education, Science, Research and Sport of the Slovak Republic, Slovakia; National Research Foundation of South Africa, South Africa; Swedish Research Council (VR) and Knut & Alice Wallenberg Foundation (KAW), Sweden; European Organization for Nuclear Research, Switzerland; Suranaree University of Technology (SUT), National Science and Technology Development Agency (NSTDA) and National Science, Research and Innovation Fund (NSRF via PMU-B B05F650021), Thailand; Turkish Energy, Nuclear and Mineral Research Agency (TENMAK), Turkey; National Academy of Sciences of Ukraine, Ukraine; Science and Technology Facilities Council (STFC), United Kingdom; National Science Foundation of the United States of America (NSF) and United States Department of Energy, Office of Nuclear Physics (DOE NP), United States of America. In addition, individual groups or members have received support from: Czech Science Foundation (grant no. 23-07499S), Czech Republic; FORTE project, reg. no. CZ.02.01.01/00/22_008/0004632, Czech Republic, co-funded by the European Union, Czech Republic; European Research Council

(grant no. 950692), European Union; ICSC—Centro Nazionale di Ricerca in High Performance Computing, Big Data and Quantum Computing, European Union - NextGenerationEU; Academy of Finland (Center of Excellence in Quark Matter) (grant nos. 346327, 346328), Finland; Deutsche Forschungs Gemeinschaft (DFG, German Research Foundation) "Neutrinos and Dark Matter in Astro- and Particle Physics" (grant no. SFB 1258), Germany.

## Author contributions

The work reported in this document is the result of the ALICE Collaboration effort.

## Competing interests

The authors declare no competing interests.

## Additional information

## The ALICE Collaboration

S. Acharya [1], A. Agarwal[2], G. Aglieri Rinella [3], L. Aglietta [4], M. Agnello [5], N. Agrawal [6], Z. Ahammed[2], S. Ahmad [7], S. U. Ahn [8], I. Ahuja [9], A. Akindinov [10], V. Akishina[11], M. Al-Turany [12], D. Aleksandrov [10], B. Alessandro [13], H. M. Alfanda [14], R. Alfaro Molina [15], B. Ali [7], A. Alici [6], N. Alizadehvandchali [16], A. Alkin [17], J. Alme [18], G. Alocco [4,19], T. Alt [20], A. R. Altamura [21], I. Altsybeev [22], J. R. Alvarado [23], C. O. R. Alvarez [23], M. N. Anaam [14], C. Andrei [24], N. Andreou [25], A. Andronic [26], E. Andronov [10], V. Anguelov [27], F. Antinori [28], P. Antonioli [29], N. Apadula [30], L. Aphecetche [31], H. Appelshäuser [20], C. Arata [32], S. Arcelli [6], R. Arnaldi [13], J. G. M. C. A. Arneiro [33], I. C. Arsene [34], M. Arslandok [35], A. Augustinus [3], R. Averbeck [12], D. Averyanov [10], M. D. Azmi [7], H. Baba[36], A. Badalà [37], J. Bae [17], Y. Bae [17], Y. W. Baek [38], X. Bai [39], R. Bailhache [20], Y. Bailung [40], R. Bala [41], A. Balbino [5], A. Baldisseri [42], B. Balis [43], Z. Banoo [41], V. Barbasova[9], F. Barile [44], L. Barioglio [13], M. Barlou[45], B. Barman[46], G. G. Barnaföldi [47], L. S. Barnby [25], E. Barreau [31], V. Barret [1], L. Barreto [33], C. Bartels [48], K. Barth [3], E. Bartsch [20], N. Bastid [1], S. Basu [49], G. Batigne [31], D. Battistini [22], B. Batyunya [50], D. Bauri[51], J. L. Bazo Alba [52], I. G. Bearden [53], C. Beattie [35], P. Becht [12], D. Behera [40], I. Belikov [54], A. D. C. Bell Hechavarria [26], F. Bellini [6], R. Bellwied [16], S. Belokurova [10], L. G. E. Beltran [55], Y. A. V. Beltran [23], G. Bencedi [47], A. Bensaoula[16], S. Beole [4], Y. Berdnikov [10], A. Berdnikova [27], L. Bergmann [27], M. G. Besoiu [56], L. Betev [3], P. P. Bhaduri [2], A. Bhasin [41], B. Bhattacharjee [46], L. Bianchi [4], J. Bielčík [57], J. Bielčková [58], A. P. Bigot [54], A. Bilandzic [22], G. Biro [47], S. Biswas [59], N. Bize [31], J. T. Blair [60], D. Blau [10], M. B. Blidaru [12], N. Bluhme[11], C. Blume [20], F. Bock [61], T. Bodova [18], J. Bok [62], L. Boldizsár [47], M. Bombara [9], P. M. Bond [3], G. Bonomi [63,64], H. Borel [42], A. Borissov [10], A. G. Borquez Carcamo [27], E. Botta [4], Y. E. M. Bouziani [20], L. Bratrud [20], P. Braun-Munzinger [12], M. Bregant [33], M. Broz [57], G. E. Bruno [44,65], V. D. Buchakchiev [66], M. D. Buckland [67], D. Budnikov [10], H. Buesching [20], S. Bufalino [5], P. Buhler [68], N. Burmasov [10], Z. Buthelezi [69,70], A. Bylinkin [18], S. A. Bysiak[71], J. C. Cabanillas Noris [55], M. F. T. Cabrera[16], H. Caines [35], A. Caliva [72], E. Calvo Villar [52], J. M. M. Camacho [55], P. Camerini [73], F. D. M. Canedo [33], S. L. Cantway [35], M. Carabas [74], A. A. Carballo [3], F. Carnesecchi [3], R. Caron [75], L. A. D. Carvalho [33], J. Castillo Castellanos [42], M. Castoldi [3], F. Catalano [3], S. Cattaruzzi [73], R. Cerri [4], I. Chakaberia [30], P. Chakraborty [76], S. Chandra [2], S. Chapeland [3], M. Chartier [48], S. Chattopadhay[2], M. Chen[77], T. Cheng [14], C. Cheshkov [75], D. Chiappara [78], V. Chibante Barroso [3], D. D. Chinellato [68], E. S. Chizzali [22,142], J. Cho [79], S. Cho [79], P. Chochula [3], Z. A. Chochulska[76], D. Choudhury[46], S. Choudhury[80], P. Christakoglou [81], C. H. Christensen [53], P. Christiansen [49], T. Chujo [82], M. Ciacco [5], C. Cicalo [19], F. Cindolo [29], M. R. Ciupek[12], G. Clai[29,143], F. Colamaria [21], J. S. Colburn[83], D. Colella [44], A. Colelli[44], M. Colocci [6], M. Concas [3], G. Conesa Balbastre [32], Z. Conesa del Valle [84], G. Contin [73], J. G. Contreras [57], M. L. Coquet [31], P. Cortese [13,85],

M. R. Cosentino [86], F. Costa [3], S. Costanza [64,87], C. Cot [84], P. Crochet [1], M. M. Czarnynoga[76], A. Dainese [28], G. Dange[11], M. C. Danisch [27], A. Danu [56], P. Das [3,88], S. Das [59], A. R. Dash [26], S. Dash [51], A. De Caro [72], G. de Cataldo [21], J. de Cuveland[11], A. De Falco [89], D. De Gruttola [72], N. De Marco [13], C. De Martin [73], S. De Pasquale [72], R. Deb [63], R. Del Grande [22], L. Dello Stritto [3], W. Deng [14], K. C. Devereaux[90], G. G. A. de Souza[33], P. Dhankher [90], D. Di Bari [44], A. Di Mauro [3], B. Di Ruzza [91], B. Diab [42], R. A. Diaz [50,92], Y. Ding [14], J. Ditzel [20], R. Divià [3], Ø. Djuvsland[18], U. Dmitrieva [10], A. Dobrin [56], B. Dönigus [20], J. M. Dubinski [76], A. Dubla [12], P. Dupieux [1], N. Dzalaiova[93], T. M. Eder [26], R. J. Ehlers [30], F. Eisenhut [20], R. Ejima [94], D. Elia [21], B. Erazmus [31], F. Ercolessi [6], B. Espagnon [84], G. Eulisse [3], D. Evans [83], S. Evdokimov [10], L. Fabbietti [22], M. Faggin [73], J. Faivre [32], F. Fan [14], W. Fan [30], A. Fantoni [95], M. Fasel [61], G. Feofilov [10], A. Fernández Téllez [23], L. Ferrandi [33], M. B. Ferrer [3], A. Ferrero [42], C. Ferrero [13,144], A. Ferretti [4], V. J. G. Feuillard [27], V. Filova [57], D. Finogeev [10], F. M. Fionda [19], E. Flatland[3], F. Flor [16,35], A. N. Flores [60], S. Foertsch [69], I. Fokin [27], S. Fokin [10], U. Follo [13,144], E. Fragiacomo [96], E. Frajna [47], U. Fuchs [3], N. Funicello [72], C. Furget [32], A. Furs [10], T. Fusayasu [97], J. J. Gaardhøje [53], M. Gagliardi [4], A. M. Gago [52], T. Gahlaut[51], C. D. Galvan [55], S. Gami[88], D. R. Gangadharan [16], P. Ganoti [45], C. Garabatos [12], J. M. Garcia [23], T. García Chávez [23], E. Garcia-Solis [98], C. Gargiulo [3], P. Gasik [12], H. M. Gaur[11], A. Gautam [99], M. B. Gay Ducati [100], M. Germain [31], R. A. Gernhaeuser[22], C. Ghosh[2], M. Giacalone [29], G. Gioachin [5], S. K. Giri[2], P. Giubellino [12,13], P. Giubilato [78], A. M. C. Glaenzer [42], P. Glässel [27], E. Glimos [101], D. J. Q. Goh[102], V. Gonzalez [103], P. Gordeev [10], M. Gorgon [43], K. Goswami [40], S. Gotovac [104], V. Grabski [15], L. K. Graczykowski [76], E. Grecka [58], A. Grelli [105], C. Grigoras [3], V. Grigoriev [10], S. Grigoryan [50,106], F. Grosa [3], J. F. Grosse-Oetringhaus [3], R. Grosso [12], D. Grund [57], N. A. Grunwald[27], G. G. Guardiano [107], R. Guernane [32], M. Guilbaud [31], K. Gulbrandsen [53], J. J. W. K. Gumprecht[68], T. Gündem [20], T. Gunji [36], W. Guo [14], A. Gupta [41], R. Gupta [41], R. Gupta [40], K. Gwizdziel [76], L. Gyulai [47], C. Hadjidakis [84], F. U. Haider [41], S. Haidlova [57], M. Haldar[59], H. Hamagaki [102], Y. Han [108], B. G. Hanley [103], R. Hannigan [60], J. Hansen [49], M. R. Haque [12], J. W. Harris [35], A. Harton [98], M. V. Hartung [20], H. Hassan [109], D. Hatzifotiadou [29], P. Hauer [110], L. B. Havener [35], E. Hellbär [3], H. Helstrup [111], M. Hemmer [20], T. Herman [57], S. G. Hernandez[16], G. Herrera Corral [112], S. Herrmann [75], K. F. Hetland [111], B. Heybeck [20], H. Hillemanns [3], B. Hippolyte [54], I. P. M. Hobus[81], F. W. Hoffmann [113], B. Hofman [105], M. Horst [22], A. Horzyk [43], Y. Hou [14], P. Hristov [3], P. Huhn[20], L. M. Huhta [109], T. J. Humanic [114], A. Hutson [16], D. Hutter [11], M. C. Hwang [90], R. Ilkaev[10], M. Inaba [82], G. M. Innocenti [3], M. Ippolitov [10], A. Isakov [81], T. Isidori [99], M. S. Islam [51,80], S. Iurchenko [10], M. Ivanov [12], M. Ivanov[93], V. Ivanov [10], K. E. Iversen [49], M. Jablonski [43], B. Jacak [30,90], N. Jacazio [6], P. M. Jacobs [30], S. Jadlovska[115], J. Jadlovsky[115], S. Jaelani [116], C. Jahnke [33], M. J. Jakubowska [76], M. A. Janik [76], T. Janson[113], S. Ji [62], S. Jia [117], T. Jiang [117], A. A. P. Jimenez [118], F. Jonas [30], D. M. Jones [48], J. M. Jowett [3,12], J. Jung [20], M. Jung [20], A. Junique [3], A. Jusko [83], J. Kaewjai[119], P. Kalinak [120], A. Kalweit [3], A. Karasu Uysal [121], D. Karatovic [122], N. Karatzenis[83], O. Karavichev [10], T. Karavicheva [10], E. Karpechev [10], M. J. Karwowska [76], U. Kebschull [113], M. Keil [3], B. Ketzer [110], J. Keul [20], S. S. Khade [40], A. M. Khan [39], S. Khan [7], A. Khanzadeev [10], Y. Kharlov [10], A. Khatun [99], A. Khuntia [57], Z. Khuranova [20], B. Kileng [111], B. Kim [17], C. Kim [62], D. J. Kim [109], D. Kim [17], E. J. Kim [123], J. Kim [108], J. Kim [79], J. Kim [3,123], M. Kim [90], S. Kim [124], T. Kim [108], K. Kimura [94], A. Kirkova[66], S. Kirsch [20], I. Kisel [11], S. Kiselev [10], A. Kisiel [76], J. L. Klay [125], J. Klein [3], S. Klein [30], C. Klein-Bösing [26], M. Kleiner [20], T. Klemenz [22], A. Kluge [3], C. Kobdaj [119], R. Kohara[36], T. Kollegger[12], A. Kondratyev [50], N. Kondratyeva [10], J. Konig [20], S. A. Konigstorfer [22], P. J. Konopka [3], G. Kornakov [76], M. Korwieser [22], S. D. Koryciak [43], C. Koster[81], A. Kotliarov [58], N. Kovacic[122], V. Kovalenko [10], M. Kowalski [71], V. Kozhuharov [66], G. Kozlov [11], I. Králik [120], A. Kravčáková [9], L. Krcal [3,11], M. Krivda [83,120], F. Krizek [58], K. Krizkova Gajdosova [3], C. Krug [100], M. Krüger [20], D. M. Krupova [57], E. Kryshen [10], V. Kučera [79], C. Kuhn [54], P. G. Kuijer [81], T. Kumaoka[82], D. Kumar[2], L. Kumar [126], N. Kumar[126], S. Kumar [21], S. Kundu [3], P. Kurashvili [127], A. B. Kurepin [10], A. Kuryakin [10], S. Kushpil [58], V. Kuskov [10], M. Kutyla[76], A. Kuznetsov [50], M. J. Kweon [79], Y. Kwon [108], S. L. La Pointe [11], P. La Rocca [128], A. Lakrathok[119], M. Lamanna [3], A. R. Landou [32], R. Langoy [129], P. Larionov [3], E. Laudi [3], L. Lautner [22], R. A. N. Laveaga[55], R. Lavicka [68], R. Lea [63,64], H. Lee [17], I. Legrand [24], G. Legras [26], J. Lehrbach [11], A. M. Lejeune[57], T. M. Lelek[43], R. C. Lemmon [67,148], I. León Monzón [55], M. M. Lesch [22], P. Lévai [47], M. Li[14], P. Li[117], X. Li[117], B. E. Liang-Gilman [90], J. Lien [129], R. Lietava [83], I. Likmeta [16], B. Lim [4], H. Lim [62], S. H. Lim [62], V. Lindenstruth [11], C. Lippmann [12], D. Liskova[115], D. H. Liu [14], J. Liu [48], G. S. S. Liveraro [107], I. M. Lofnes [18], C. Loizides [61], S. Lokos [71], J. Lömker [105], X. Lopez [1], E. López Torres [92], C. Lotteau[75], P. Lu [12,39], Z. Lu [117], F. V. Lugo [15], J. R. Luhder [26], G. Luparello [96], Y. G. Ma [77], M. Mager [3], A. Maire [54], E. M. Majerz [43], M. V. Makariev [66], M. Malaev [10], G. Malfattore [6], N. M. Malik [41], S. K. Malik [41], D. Mallick [84], N. Mallick [40,109], G. Mandaglio [37,130], S. K. Mandal [127], A. Manea [56], V. Manko [10], F. Manso [1], V. Manzari [21], Y. Mao [14], R. W. Marcjan [43], G. V. Margagliotti [73], A. Margotti [29], A. Marin [12], C. Markert [60], C. F. B. Marquez[44], P. Martinengo [3],

M. I. Martinez [23], G. Martinez Garcia [31], M. P. P. Martins [33], S. Masciocchi [12], M. Masera [4], A. Masoni [19], L. Massacrier [84], O. Massen [105], A. Mastroserio [21,91], S. Mattiazzo [78], A. Matyja [71], F. Mazzaschi [3,4], M. Mazzilli [16], Y. Melikyan [131], M. Melo [33], A. Menchaca-Rocha [15], J. E. M. Mendez [118], E. Meninno [68], A. S. Menon [16], M. W. Menzel [3,27], M. Meres [93], L. Micheletti [3], D. Mihai [74], D. L. Mihaylov [22], K. Mikhaylov [10,50], N. Minafra [99], D. Miśkowiec [12], A. Modak [63], B. Mohanty [88], M. Mohisin Khan [7,145], M. A. Molander [131], M. M. Mondal [88], S. Monira [76], C. Mordasini [109], D. A. Moreira De Godoy [26], I. Morozov [10], A. Morsch [3], T. Mrnjavac [3], V. Muccifora [95], S. Muhuri [2], J. D. Mulligan [30], A. Mulliri [89], M. G. Munhoz [33], R. H. Munzer [20], H. Murakami [36], S. Murray [132], L. Musa [3], J. Musinsky [120], J. W. Myrcha [76], B. Naik [70], A. I. Nambrath [90], B. K. Nandi [51], R. Nania [29], E. Nappi [21], A. F. Nassirpour [124], V. Nastase [74], A. Nath [27], S. Nath [2], C. Nattrass [101], M. N. Naydenov [66], A. Neagu [34], A. Negru [74], E. Nekrasova [10], L. Nellen [118], R. Nepeivoda [49], S. Nese [34], N. Nicassio [44], B. S. Nielsen [53], E. G. Nielsen [53], S. Nikolaev [10], V. Nikulin [10], F. Noferini [29], S. Noh [133], P. Nomokonov [50], J. Norman [48], N. Novitzky [61], P. Nowakowski [76], A. Nyanin [10], J. Nystrand [18], S. Oh [124], A. Ohlson [49], V. A. Okorokov [10], J. Oleniacz [76], A. Onnerstad [109], C. Oppedisano [13], A. Ortiz Velasquez [118], J. Otwinowski [71], M. Oya [94], K. Oyama [102], S. Padhan [51], D. Pagano [63,64], G. Paić [118], S. Paisano-Guzmán [23], A. Palasciano [21], I. Panasenko [49], S. Panebianco [42], C. Pantouvakis [78], H. Park [82], J. Park [82], S. Park [17], J. E. Parkkila [3], Y. Patley [51], R. N. Patra [21], B. Paul [2], H. Pei [14], T. Peitzmann [105], X. Peng [134], M. Pennisi [4], S. Perciballi [4], D. Peresunko [10], G. M. Perez [92], Y. Pestov [10], M. T. Petersen [53], V. Petrov [10], M. Petrovici [24], S. Piano [96], M. Pikna [93], P. Pillot [31], O. Pinazza [3,29], L. Pinsky [16], C. Pinto [22], S. Pisano [95], M. Płoskoń [30], M. Planinic [122], D. K. Plociennik [43], M. G. Poghosyan [61], B. Polichtchouk [10], S. Politano [5], N. Poljak [122], A. Pop [24], S. Porteboeuf-Houssais [1], V. Pozdniakov [50,149], I. Y. Pozos [23], K. K. Pradhan [40], S. K. Prasad [59], S. Prasad [40], R. Preghenella [29], F. Prino [13], C. A. Pruneau [103], I. Pshenichnov [10], M. Puccio [3], S. Pucillo [4], S. Qiu [81], L. Quaglia [4], A. M. K. Radhakrishnan [40], S. Ragoni [135], A. Rai [35], A. Rakotozafindrabe [42], L. Ramello [13,85], M. Rasa [128], S. S. Räsänen [131], R. Rath [29], M. P. Rauch [18], I. Ravasenga [3], K. F. Read [61,101], C. Reckziegel [86], A. R. Redelbach [11], K. Redlich [127,146], C. A. Reetz [12], H. D. Regules-Medel [23], A. Rehman [18], F. Reidt [3], H. A. Reme-Ness [111], K. Reygers [27], A. Riabov [10], V. Riabov [10], R. Ricci [72], M. Richter [18], A. A. Riedel [22], W. Riegler [3], A. G. Riffero [4], M. Rignanese [78], C. Ripoli [72], C. Ristea [56], M. V. Rodriguez [3], M. Rodríguez Cahuantzi [23], S. A. Rodríguez Ramírez [23], K. Røed [34], R. Rogalev [10], E. Rogochaya [50], T. S. Rogoschinski [20], D. Rohr [3], D. Röhrich [18], S. Rojas Torres [57], P. S. Rokita [76], G. Romanenko [6], F. Ronchetti [3], E. D. Rosas [118], K. Roslon [76], A. Rossi [28], A. Roy [40], S. Roy [51], N. Rubini [6,29], J. A. Rudolph [81], D. Ruggiano [76], R. Rui [73], P. G. Russek [43], R. Russo [81], A. Rustamov [136], E. Ryabinkin [10], Y. Ryabov [10], A. Rybicki [71], J. Ryu [62], W. Rzesa [76], B. Sabiu [29], S. Sadovsky [10], J. Saetre [18], S. Saha [88], B. Sahoo [40], R. Sahoo [40], S. Sahoo [137], D. Sahu [40], P. K. Sahu [137], J. Saini [2], K. Sajdakova [9], S. Sakai [82], M. P. Salvan [12], S. Sambyal [41], D. Samitz [68], I. Sanna [3,22], T. B. Saramela [33], D. Sarkar [51,53], P. Sarma [46], V. Sarritzu [89], V. M. Sarti [22], M. H. P. Sas [3], S. Sawan [88], E. Scapparone [29], J. Schambach [61], H. S. Scheid [20], C. Schiaua [24], R. Schicker [27], F. Schlepper [27], A. Schmah [12], C. Schmidt [12], M. O. Schmidt [3], M. Schmidt [138], N. V. Schmidt [61], A. R. Schmier [101], J. Schoengarth [20], R. Schotter [54,68], A. Schröter [11], J. Schukraft [3], K. Schweda [12], G. Scioli [6], E. Scomparin [13], J. E. Seger [135], Y. Sekiguchi [36], D. Sekihata [36], M. Selina [81], I. Selyuzhenkov [12], S. Senyukov [54], J. J. Seo [27], D. Serebryakov [10], L. Serkin [118,147], L. Šerkšnytė [22], A. Sevcenco [56], T. J. Shaba [69], A. Shabetai [31], R. Shahoyan [3], A. Shangaraev [10], B. Sharma [41], D. Sharma [51], H. Sharma [28], M. Sharma [41], S. Sharma [102], S. Sharma [41], U. Sharma [41], A. Shatat [84], O. Sheibani [16,103], K. Shigaki [94], M. Shimomura [139], J. Shin [133], S. Shirinkin [10], Q. Shou [77], Y. Sibiriak [10], S. Siddhanta [19], T. Siemiarczuk [127], T. F. Silva [33], D. Silvermyr [49], T. Simantathammakul [119], R. Simeonov [66], B. Singh [41], B. Singh [22], K. Singh [40], R. Singh [88], R. Singh [41], R. Singh [12,28], S. Singh [7], V. K. Singh [2], V. Singhal [2], T. Sinha [80], B. Sitar [93], M. Sitta [13,85], T. B. Skaali [34], G. Skorodumovs [27], N. Smirnov [35], R. J. M. Snellings [105], E. H. Solheim [34], C. Sonnabend [3,12], J. M. Sonneveld [81], F. Soramel [78], A. B. Soto-Hernandez [114], R. Spijkers [81], I. Sputowska [71], J. Staa [49], J. Stachel [27], I. Stan [56], P. J. Steffanic [101], T. Stellhorn [26], S. F. Stiefelmaier [27], D. Stocco [31], I. Storehaug [34], N. J. Strangmann [20], P. Stratmann [26], S. Strazzi [6], A. Sturniolo [37,130], C. P. Stylianidis [81], A. A. P. Suaide [33], C. Suire [84], A. Suiu [3,74], M. Sukhanov [10], M. Suljic [3], R. Sultanov [10], V. Sumberia [41], S. Sumowidagdo [116], L. H. Tabares [92], S. F. Taghavi [22], J. Takahashi [107], G. J. Tambave [88], S. Tang [14], Z. Tang [39], J. D. Tapia Takaki [99], N. Tapus [74], L. A. Tarasovicova [9], M. G. Tarzila [24], A. Tauro [3], A. Tavira García [84], G. Tejeda Muñoz [23], L. Terlizzi [4], C. Terrevoli [21], S. Thakur [59], M. Thogersen [34], D. Thomas [60], A. Tikhonov [10], N. Tiltmann [3,26], A. R. Timmins [16], M. Tkacik [115], T. Tkacik [115], A. Toia [20], R. Tokumoto [94], S. Tomassini [6], K. Tomohiro [94], N. Topilskaya [10], M. Toppi [95], V. V. Torres [31], A. G. Torres Ramos [44], A. Trifiró [37,130], T. Triloki [65], A. S. Triolo [3,37,130], S. Tripathy [3], T. Tripathy [1,51], S. Trogolo [4], V. Trubnikov [140], W. H. Trzaska [109], T. P. Trzcinski [76], C. Tsolanta [34], R. Tu [77], A. Tumkin [10], R. Turrisi [28], T. S. Tveter [34], K. Ullaland [18], B. Ulukutlu [22],

S. Upadhyaya [71], A. Uras [75], G. L. Usai [89], M. Vala[9], N. Valle [64], L. V. R. van Doremalen[105], M. van Leeuwen [81], C. A. van Veen [27], R. J. G. van Weelden [81], P. Vande Vyvre [3], D. Varga [47], Z. Varga [35,47], P. Vargas Torres[118], M. Vasileiou [45], A. Vasiliev [10,150], O. Vázquez Doce [95], O. Vazquez Rueda [16], V. Vechernin [10], E. Vercellin [4], R. Verma [51], R. Vértesi [47], M. Verweij [105], L. Vickovic[104], Z. Vilakazi[70], O. Villalobos Baillie [83], A. Villani [73], A. Vinogradov [10], T. Virgili [72], M. M. O. Virta [109], A. Vodopyanov [50], B. Volkel [3], M. A. Völkl [27], S. A. Voloshin [103], G. Volpe [44], B. von Haller [3], I. Vorobyev [3], N. Vozniuk [10], J. Vláková [9], J. Wan[77], C. Wang [77], D. Wang[77], Y. Wang [77], Y. Wang [14], Z. Wang [77], A. Wegrzynek [3], F. T. Weiglhofer[11], S. C. Wenzel [3], J. P. Wessels [26], P. K. Wiacek [43], J. Wiechula [20], J. Wikne [34], G. Wilk [127], J. Wilkinson [12], G. A. Willems [26], B. Windelband [27], M. Winn [42], J. R. Wright [60], W. Wu[77], Y. Wu [39], Z. Xiong[39], R. Xu [14], A. Yadav [110], A. K. Yadav [2], Y. Yamaguchi [94], S. Yang[18], S. Yano [94], E. R. Yeats[90], Z. Yin [14], I.-K. Yoo [62], J. H. Yoon [79], H. Yu[133], S. Yuan[18], A. Yuncu [27], V. Zaccolo [73], C. Zampolli [3], F. Zanone [27], N. Zardoshti [3], A. Zarochentsev [10], P. Závada [141], N. Zaviyalov[10], M. Zhalov [10], B. Zhang [14,27], C. Zhang [42], L. Zhang [77], M. Zhang [1,14], M. Zhang [14], S. Zhang [77], X. Zhang [14], Y. Zhang[39], Z. Zhang [14], M. Zhao [117], V. Zherebchevskii [10], Y. Zhi[117], D. Zhou [14], Y. Zhou [53], J. Zhu [14,28], S. Zhu[39], Y. Zhu[14], S. C. Zugravel [13] & N. Zurlo [63,64]

[1]Université Clermont Auvergne, CNRS/IN2P3, LPC, Clermont-Ferrand, France. [1]Université Clermont Auvergne, CNRS/IN2P3, LPC, Clermont-Ferrand, France. [2]Variable Energy Cyclotron Centre, Homi Bhabha National Institute, Kolkata, India. [3]European Organization for Nuclear Research (CERN), Geneva, Switzerland. [4]Dipartimento di Fisica dell'Università and Sezione INFN, Turin, Italy. [5]Dipartimento DISAT del Politecnico and Sezione INFN, Turin, Italy. [6]Dipartimento di Fisica e Astronomia dell'Università and Sezione INFN, Bologna, Italy. [7]Department of Physics, Aligarh Muslim University, Aligarh, India. [8]Korea Institute of Science and Technology Information, Daejeon, Republic of Korea. [9]Faculty of Science, P.J. Šafárik University, Košice, Slovak Republic. [10]Affiliated with an institute covered by a cooperation agreement with CERN, Geneva, Switzerland. [11]Frankfurt Institute for Advanced Studies, Johann Wolfgang Goethe-Universität Frankfurt, Frankfurt, Germany. [12]Research Division and ExtreMe Matter Institute EMMI, GSI Helmholtzzentrum für Schwerionenforschung GmbH, Darmstadt, Germany. [13]INFN, Sezione di Torino, Turin, Italy. [14]Central China Normal University, Wuhan, China. [15]Instituto de Fsica, Universidad Nacional Autónoma de México, Mexico City, Mexico. [16]University of Houston, Houston, TX, USA. [17]Sungkyunkwan University, Suwon City, Republic of Korea. [18]Department of Physics and Technology, University of Bergen, Bergen, Norway. [19]INFN, Sezione di Cagliari, Cagliari, Italy. [20]Institut für Kernphysik, Johann Wolfgang Goethe-Universität Frankfurt, Frankfurt, Germany. [21]INFN, Sezione di Bari, Bari, Italy. [22]Physik Department, Technische Universität München, Munich, Germany. [23]High Energy Physics Group, Universidad Autónoma de Puebla, Puebla, Mexico. [24]Horia Hulubei National Institute of Physics and Nuclear Engineering, Bucharest, Romania. [25]University of Derby, Derby, UK. [26]Universität Münster, Institut für Kernphysik, Münster, Germany. [27]Physikalisches Institut, Ruprecht-Karls-Universität Heidelberg, Heidelberg, Germany. [28]INFN, Sezione di Padova, Padova, Italy. [29]INFN, Sezione di Bologna, Bologna, Italy. [30]Lawrence Berkeley National Laboratory, Berkeley, CA, USA. [31]SUBATECH, IMT Atlantique, Nantes Université, CNRS-IN2P3, Nantes, France. [32]Laboratoire de Physique Subatomique et de Cosmologie, Université Grenoble-Alpes, CNRS-IN2P3, Grenoble, France. [33]Universidade de São Paulo (USP), São Paulo, Brazil. [34]Department of Physics, University of Oslo, Oslo, Norway. [35]Yale University, New Haven, CT, USA. [36]University of Tokyo, Tokyo, Japan. [37]INFN, Sezione di Catania, Catania, Italy. [38]Gangneung-Wonju National University, Gangneung, Republic of Korea. [39]University of Science and Technology of China, Hefei, China. [40]Indian Institute of Technology Indore, Indore, India. [41]Physics Department, University of Jammu, Jammu, India. [42]Université Paris-Saclay, Centre d'Etudes de Saclay (CEA), IRFU, Département de Physique Nucléaire (DPhN), Saclay, France. [43]AGH University of Krakow, Cracow, Poland. [44]Dipartimento Interateneo di Fisica 'M. Merlin' and Sezione INFN, Bari, Italy. [45]Department of Physics, National and Kapodistrian University of Athens, School of Science, Athens, Greece. [46]Department of Physics, Gauhati University, Guwahati, India. [47]HUN-REN Wigner Research Centre for Physics, Budapest, Hungary. [48]University of Liverpool, Liverpool, UK. [49]Division of Particle Physics, Lund University Department of Physics, Lund, Sweden. [50]Affiliated with an international laboratory covered by a cooperation agreement with CERN, Geneva, Switzerland. [51]Indian Institute of Technology Bombay (IIT), Mumbai, India. [52]Sección Fsica, Departamento de Ciencias, Pontificia Universidad Católica del Perú, Lima, Peru. [53]Niels Bohr Institute, University of Copenhagen, Copenhagen, Denmark. [54]Université de Strasbourg, CNRS, IPHC, Strasbourg, France. [55]Universidad Autónoma de Sinaloa, Culiacán, Mexico. [56]Institute of Space Science (ISS), Bucharest, Romania. [57]Faculty of Nuclear Sciences and Physical Engineering, Czech Technical University in Prague, Prague, Czech Republic. [58]Nuclear Physics Institute of the Czech Academy of Sciences, Husinec-Řež, Czech Republic. [59]Department of Physics and Centre for Astroparticle Physics and Space Science (CAPSS), Bose Institute, Kolkata, India. [60]The University of Texas at Austin, Austin, TX, USA. [61]Oak Ridge National Laboratory, Oak Ridge, TN, USA. [62]Department of Physics, Pusan National University, Pusan, Republic of Korea. [63]Università di Brescia, Brescia, Italy. [64]INFN, Sezione di Pavia, Pavia, Italy. [65]Politecnico di Bari and Sezione INFN, Bari, Italy. [66]Faculty of Physics, Sofia University, Sofia, Bulgaria. [67]Nuclear Physics Group, STFC Daresbury Laboratory, Daresbury, UK. [68]Stefan Meyer Institut für Subatomare Physik (SMI), Vienna, Austria. [69]iThemba LABS, National Research Foundation, Somerset West, South Africa. [70]University of the Witwatersrand, Johannesburg, South Africa. [71]The Henryk Niewodniczanski Institute of Nuclear Physics, Polish Academy of Sciences, Cracow, Poland. [72]Dipartimento di Fisica 'E.R. Caianiello' dell'Università and Gruppo Collegato INFN, Salerno, Italy. [73]Dipartimento di Fisica dell'Università and Sezione INFN, Trieste, Italy. [74]Universitatea Nationala de Stiinta si Tehnologie Politehnica Bucuresti, Bucharest, Romania. [75]Institut de Physique des 2 Infinis de Lyon, Université de Lyon, CNRS/IN2P3, Lyon, France. [76]Warsaw University of Technology, Warsaw, Poland. [77]Fudan University, Shanghai, China. [78]Dipartimento di Fisica e Astronomia dell'Università and Sezione INFN, Padova, Italy. [79]Inha University, Incheon, Republic of Korea. [80]Saha Institute of Nuclear Physics, Homi Bhabha National Institute, Kolkata, India. [81]Nikhef, National Institute for Subatomic Physics, Amsterdam, Netherlands. [82]University of Tsukuba, Tsukuba, Japan. [83]School of Physics and Astronomy, University of Birmingham, Birmingham, UK. [84]Université Paris-Saclay, CNRS/IN2P3, IJCLab, Orsay, France. [85]Università del Piemonte Orientale, Vercelli, Italy. [86]Universidade Federal do ABC, Santo Andre, Brazil. [87]Dipartimento di Fisica, Università di Pavia, Pavia, Italy. [88]National Institute of Science Education and Research, Homi Bhabha National Institute, Jatni, India. [89]Dipartimento di Fisica dell'Università and Sezione INFN, Cagliari, Italy. [90]Department of Physics, University of California, Berkeley, CA, USA. [91]Università degli Studi di Foggia, Foggia, Italy. [92]Centro de Aplicaciones Tecnológicas y Desarrollo Nuclear (CEADEN), Havana, Cuba. [93]Comenius University Bratislava, Faculty of Mathematics, Physics and Informatics, Bratislava, Slovak Republic. [94]Physics Program and International Institute for Sustainability with Knotted Chiral Meta Matter (WPI-SKCM²), Hiroshima University, Hiroshima, Japan. [95]INFN, Laboratori Nazionali di Frascati, Frascati, Italy. [96]INFN, Sezione di Trieste, Trieste, Italy. [97]Saga University, Saga, Japan. [98]Chicago State University, Chicago, IL, USA. [99]University of Kansas, Lawrence, KS, USA. [100]Instituto de Física, Universidade Federal do Rio Grande do Sul (UFRGS), Porto Alegre, Brazil. [101]University of Tennessee, Knoxville, TN, USA. [102]Nagasaki Institute of Applied Science, Nagasaki, Japan.

[103]Wayne State University, Detroit, MI, USA. [104]Faculty of Electrical Engineering, Mechanical Engineering and Naval Architecture, University of Split, Split, Croatia. [105]Institute for Gravitational and Subatomic Physics (GRASP), Utrecht University/Nikhef, Utrecht, Netherlands. [106]A.I. Alikhanyan National Science Laboratory (Yerevan Physics Institute) Foundation, Yerevan, Armenia. [107]Universidade Estadual de Campinas (UNICAMP), Campinas, Brazil. [108]Yonsei University, Seoul, Republic of Korea. [109]University of Jyväskylä, Jyväskylä, Finland. [110]Helmholtz-Institut für Strahlen- und Kernphysik, Rheinische Friedrich-Wilhelms-Universität Bonn, Bonn, Germany. [111]Faculty of Technology, Environmental and Social Sciences, Bergen, Norway. [112]Centro de Investigación y de Estudios Avanzados (CINVESTAV), Mexico City and Mérida, Mexico City, Mexico. [113]Johann-Wolfgang-Goethe Universität Frankfurt Institut für Informatik, Fachbereich Informatik und Mathematik, Frankfurt, Germany. [114]Ohio State University, Ohio, OH, USA. [115]Technical University of Košice, Košice, Slovak Republic. [116]National Research and Innovation Agency—BRIN, Jakarta, Indonesia. [117]China Institute of Atomic Energy, Beijing, China. [118]Instituto de Ciencias Nucleares, Universidad Nacional Autónoma de México, Mexico City, Mexico. [119]Suranaree University of Technology, Nakhon Ratchasima, Thailand. [120]Institute of Experimental Physics, Slovak Academy of Sciences, Košice, Slovak Republic. [121]Yildiz Technical University, Istanbul, Turkey. [122]Physics department, Faculty of science, University of Zagreb, Zagreb, Croatia. [123]Jeonbuk National University, Jeonju, Republic of Korea. [124]Department of Physics, Sejong University, Seoul, Republic of Korea. [125]California Polytechnic State University, San Luis Obispo, CA, USA. [126]Physics Department, Panjab University, Chandigarh, India. [127]National Centre for Nuclear Research, Warsaw, Poland. [128]Dipartimento di Fisica e Astronomia dell'Università and Sezione INFN, Catania, Italy. [129]University of South-Eastern Norway, Kongsberg, Norway. [130]Dipartimento di Scienze MIFT, Università di Messina, Messina, Italy. [131]Helsinki Institute of Physics (HIP), Helsinki, Finland. [132]University of Cape Town, Cape Town, South Africa. [133]Chungbuk National University, Cheongju, Republic of Korea. [134]China University of Geosciences, Wuhan, China. [135]Creighton University, Omaha, NE, USA. [136]National Nuclear Research Center, Baku, Azerbaijan. [137]Institute of Physics, Homi Bhabha National Institute, Bhubaneswar, India. [138]Physikalisches Institut, Eberhard-Karls-Universität Tübingen, Tübingen, Germany. [139]Nara Women's University (NWU), Nara, Japan. [140]Bogolyubov Institute for Theoretical Physics, National Academy of Sciences of Ukraine, Kiev, Ukraine. [141]Institute of Physics of the Czech Academy of Sciences, Prague, Czech Republic. [142]Max-Planck-Institut fur Physik, Munich, Germany. [143]Italian National Agency for New Technologies, Energy and Sustainable Economic Development (ENEA), Bologna, Italy. [144]Dipartimento DET del Politecnico di Torino, Turin, Italy. [145]Department of Applied Physics, Aligarh Muslim University, Aligarh, India. [146]Institute of Theoretical Physics, University of Wroclaw, Wroclaw, Poland. [147]Facultad de Ciencias, Universidad Nacional Autónoma de México, Mexico City, Mexico. [148]Deceased: R. C. Lemmon. [149]Deceased: V. Pozdniakov. [150]Deceased: A. Vasiliev.

