## [Transparent Peer Review file · Nature Communications]

Observation of partonic flow in proton--proton and proton--nucleus collisions

Corresponding Author: Dr Alice Publications

Version 0:

Reviewer comments:

Reviewer #1

(Remarks to the Author)

I thank the authors for their detailed responses to my comments. My primary concerns were related to the distinct nature of the physics claims over those allowed by earlier measurements, the usage of models to draw the main conclusion, and the possibility of other approaches (such as the NCQ scaling). I am convinced by the response of the authors on these points, and by the modifications of the manuscript to address them. I believe it is now appropriate to publish the manuscript in Nature Communications.

(Remarks on code availability)

Reviewer #2

(Remarks to the Author)

I thank the authors for their careful and thorough responses to my previous comments. I appreciate the significant effort that has gone into revising the manuscript, as well as the clear explanations and additional comparisons that directly address the points I raised in my initial report.

The responses provided are, in my view, satisfactory.

The manuscript now gives a much clearer account of the relationship between these new ALICE results and previous measurements from CMS, ALICE, and PHENIX. The authors correctly point out that, while some signatures of baryon--meson v_2 grouping and splitting have been reported before, their current analysis offers important advances in terms of non-flow subtraction, particle species coverage, and statistical power. These improvements make the present results the most robust to date, and the additional discussion about the limitations of earlier methods is both appropriate and helpful for the broader readership.

That said, the suppression of non-flow effects in small systems remains an open and actively debated topic within the heavy-ion community. While the authors provide a strong case that their methodology reduces non-flow contributions, it would be valuable for both the field and the readers if the manuscript were to expand the discussion of this issue further. In particular, it would be beneficial for the authors to explicitly compare their non-flow suppression strategy to those employed in previous experiments and analyses, highlighting why and how the method used in this work constitutes a substantive improvement. Such a discussion clarifies why the present results should be considered more robust than earlier measurements and provides a useful perspective for future analyses.

I recommend the paper for publication in its current form. In the manuscript, I encourage the authors to reiterate the outstanding questions regarding non-flow in small systems and to clearly articulate the evidence supporting the efficacy and relative advantages of their non-flow subtraction procedure. Such transparency will further strengthen the manuscript and help guide future work in this rapidly evolving area.

(Remarks on code availability)

Reviewer #3

(Remarks to the Author)

This work provides a first clear observation of meson-baryon v_2 groupings at intermediate p_T in high multiplicity pPb and pp collisions, which indicate the existence of partonic flow in these small collision systems. This topic should have a broad interest and is therefore suitable for publication in Nature Communications. In my earlier report to the previous version of this manuscript submitted to Nature Physics, I raised questions on the model dependence of the conclusion, reliability of the models used in this study, NCQ scaling of the hadron v_2 , and the lower limit of multiplicity when partonic flow still exists, etc. The authors have provided satisfactory responses to most of these questions and improved the manuscript accordingly. Below I have two more suggestions.

(1) While I appreciate that the authors include an AMPT calculation in the Appendix, this does not fully address my concern on the model dependence of the conclusion. The conclusion of this work is "partonic flow has been observed in pPb and pp collisions" and it is suggested that the meson-baryon v_2 groupings result from partonic flow + parton coalescence. However, the AMPT, which includes both partonic flow and parton coalescence, cannot even qualitatively show the meson-baryon v_2 grouping behavior at intermediate p_T . One can reasonably question whether this grouping behavior still be viewed as smoking-gun evidence of partonic flow? Based on the current comparisons between models and data, it seems that meson-baryon grouping requires not just partonic flow, but very strong (or hydrodynamics-like) partonic flow. I suggest the authors make statements more accurate throughout the manuscript. Since only one model can describe the data, explicitly mentioning the model dependence of the conclusion is also desired.

(2) Since more than one referee raised the question about why not use NCQ scaling as more direct evidence of the partonic degree of freedom, I suggest the authors add a brief discussion about it in the main body of the manuscript. This could also be a common question from the readers.

(Remarks on code availability)

Nature Communications referee comments:

Reviewer #1 (Remarks to the Author):

I thank the authors for their detailed responses to my comments. My primary concerns were related to the distinct nature of the physics claims over those allowed by earlier measurements, the usage of models to draw the main conclusion, and the possibility of other approaches (such as the NCQ scaling). I am convinced by the response of the authors on these points, and by the modifications of the manuscript to address them. I believe it is now appropriate to publish the manuscript in Nature Communications.

→ We sincerely thank the referee for the thoughtful comments and excellent suggestions, which have significantly improved the quality and readability of the manuscript.

Nature Communications referee comments:

Reviewer #2 (Remarks to the Author):

I thank the authors for their careful and thorough responses to my previous comments. I appreciate the significant effort that has gone into revising the manuscript, as well as the clear explanations and additional comparisons that directly address the points I raised in my initial report.

→ We sincerely thank the referee for the thoughtful comments and excellent suggestions, which have significantly improved the quality and readability of the manuscript.

The responses provided are, in my view, satisfactory.

The manuscript now gives a much clearer account of the relationship between these new ALICE results and previous measurements from CMS, ALICE, and PHENIX. The authors correctly point out that, while some signatures of baryon–meson v_2 grouping and splitting have been reported before, their current analysis offers important advances in terms of non-flow subtraction, particle species coverage, and statistical power. These improvements make the present results the most robust to date, and the additional discussion about the limitations of earlier methods is both appropriate and helpful for the broader readership.

That said, the suppression of non-flow effects in small systems remains an open and actively debated topic within the heavy-ion community. While the authors provide a strong case that their methodology reduces non-flow contributions, it would be valuable for both the field and the readers if the manuscript were to expand the discussion of this issue further. In particular, it would be beneficial for the authors to explicitly compare their non-flow suppression strategy to those employed in previous experiments and analyses, highlighting why and how the method used in this work constitutes a substantive improvement. Such a discussion clarifies why the present results should be considered more robust than earlier measurements and provides a useful perspective for future analyses.

I recommend the paper for publication in its current form. In the manuscript, I encourage the authors to reiterate the outstanding questions regarding non-flow in small systems and to clearly articulate the evidence supporting the efficacy and relative advantages of their non-flow subtraction procedure. Such transparency will further strengthen the manuscript and help guide future work in this rapidly evolving area.

→ Many thanks for your comment. We have included a discussion on non-flow subtraction in the supplementary material (Section A.4)."

"In this work, all available non-flow suppression methods (low-multiplicity subtraction~\cite{ALICE:2013snk, CMS:2018loe, CMS:2016fnw}, template fit~\cite{ATLAS:2015hzw}, and improved template fit~\cite{ATLAS:2018ngv}) have been tested,

and the residual non-flow has been estimated using PYTHIA8 for each method. Among these, the template fit provides the most effective non-flow subtraction, yielding the lowest residual non-flow ($\sim 5-7\%$) across the considered kinematic range. This residual non-flow has been included in the systematic uncertainties. The inclusion of the remaining non-flow enables better comparisons with theoretical models and supports robust, data-driven physics conclusions.”

Nature Communications referee comments:

Reviewer #3 (Remarks to the Author):

This work provides a first clear observation of meson-baryon v_2 groupings at intermediate p_T in high multiplicity pPb and pp collisions, which indicate the existence of partonic flow in these small collision systems. This topic should have a broad interest and is therefore suitable for publication in Nature Communications. In my earlier report to the previous version of this manuscript submitted to Nature Physics, I raised questions on the model dependence of the conclusion, reliability of the models used in this study, NCQ scaling of the hadron v_2 , and the lower limit of multiplicity when partonic flow still exists, etc. The authors have provided satisfactory responses to most of these questions and improved the manuscript accordingly. Below I have two more suggestions.

→ We sincerely thank the referee for the thoughtful comments and excellent suggestions, which have significantly improved the quality and readability of the manuscript.

(1) While I appreciate that the authors include an AMPT calculation in the Appendix, this does not fully address my concern on the model dependence of the conclusion. The conclusion of this work is “partonic flow has been observed in pPb and pp collisions” and it is suggested that the meson-baryon v_2 groupings result from partonic flow + parton coalescence. However, the AMPT, which includes both partonic flow and parton coalescence, cannot even qualitatively show the meson-baryon v_2 grouping behavior at intermediate p_T . One can reasonably question whether this grouping behavior still be viewed as smoking-gun evidence of partonic flow? Based on the current comparisons between models and data, it seems that meson-baryon grouping requires not just partonic flow, but very strong (or hydrodynamics-like) partonic flow. I suggest the authors make statements more accurate throughout the manuscript. Since only one model can describe the data, explicitly mentioning the model dependence of the conclusion is also desired.

→ Thank you for your comment. We have explicitly mentioned 'hydrodynamic flow of partons' in two places in the updated draft:

L 176: *“Therefore, these results provide evidence of hadronization via coalescence of hydrodynamically flowing quarks in small collision systems at the LHC.”*

L 204: *“ The presented measurements and the comparisons with available theoretical model calculations provide evidence that the system created in high-multiplicity p--Pb and pp collisions*

includes a stage with hydrodynamically flowing partons, similar to the one observed in heavy-ion collisions.”

(2) Since more than one referee raised the question about why not use NCQ scaling as more direct evidence of the partonic degree of freedom, I suggest the authors add a brief discussion about it in the main body of the manuscript. This could also be a common question from the readers.

→ Thank you for the excellent suggestions. We have included a discussion on NCQ scaling in the main body of the manuscript.”

“In Refs.~\cite{CMS:2014und, CMS:2018loe}, the number-of-constituent-quark (NCQ) scaling of v_2 has also been studied. This scaling was initially attributed to hadron production via the coalescence of thermal partons in heavy-ion collisions~\cite{Fries:2008hs, Fries:2003vb, Molnar:2003ff}. However, advanced coalescence models incorporate the recombination of thermal quarks with shower quarks originating from jet-medium interactions to describe the v_2 spectra and v_2 of identified particles over a broad p_T range~\cite{Zhao:2021vmu}, differing from the coalescence mechanism~\cite{Molnar:2003ff, Fries:2008hs, Fries:2003vb} associated with NCQ scaling of v_2 . In addition, contributions from radial flow and jet fragmentation at intermediate p_T can also lead to deviations from NCQ scaling. Notably, the ALICE measurements exhibit deviations from NCQ scaling at the level of $\pm 20\%$ in Pb--Pb collisions~\cite{ALICE:2014wao, ALICE:2022zks}. This underscores the need for a better understanding of this scaling as evidence of partonic collectivity in relativistic collisions.”